# Gating the pore of the calcium-activated chloride channel TMEM16A

Andy K. M. Lam [1✉], Jan Rheinberger [2], Cristina Paulino [2✉] & Raimund Dutzler [1✉]

The binding of cytoplasmic $Ca^{2+}$ to the anion-selective channel TMEM16A triggers a conformational change around its binding site that is coupled to the release of a gate at the constricted neck of an hourglass-shaped pore. By combining mutagenesis, electrophysiology, and cryo-electron microscopy, we identified three hydrophobic residues at the intracellular entrance of the neck as constituents of this gate. Mutation of each of these residues increases the potency of $Ca^{2+}$ and results in pronounced basal activity. The structure of an activating mutant shows a conformational change of an α-helix that contributes to $Ca^{2+}$ binding as a likely cause for the basal activity. Although not in physical contact, the three residues are functionally coupled to collectively contribute to the stabilization of the gate in the closed conformation of the pore, thus explaining the low open probability of the channel in the absence of $Ca^{2+}$.

[1] Department of Biochemistry, University of Zurich, Winterthurerstrasse 190, CH-8057 Zurich, Switzerland. [2] Department of Structural Biology and Membrane Enzymology at the Groningen Biomolecular Sciences and Biotechnology Institute, University of Groningen, Nijenborgh 4, 9747 AG Groningen, The Netherlands. ✉email: a.lam@bioc.uzh.ch; c.paulino@rug.nl; dutzler@bioc.uzh.ch

Calcium-activated chloride channels (CACC) facilitate transmembrane anion conduction in response to the increase of the intracellular $Ca^{2+}$ concentration[1]. These proteins are involved in diverse physiological processes ranging from electrical signaling to epithelial transport. The most prominent CACC is formed by TMEM16A, which is expressed in different tissues of the human body[2–4]. Whereas in endothelial smooth muscle cells, activation of TMEM16A increases their electrical excitability[5], in airway epithelia the protein contributes to chloride secretion, which makes it a promising pharmaceutical target for the treatment of cystic fibrosis[6,7].

TMEM16A is a member of the TMEM16 family of eukaryotic membrane proteins, which comprise ion channels and lipid scramblases with a conserved molecular scaffold[8,9]. Structures of both functional branches have revealed the general architecture of the family[10–14]. These proteins form homodimers with subunits containing ten membrane-spanning segments. In TMEM16 scramblases, the region involved in lipid conduction is contained within each subunit and consists of a membrane-spanning hydrophilic furrow that accommodates polar lipid headgroups during their translocation between the inner and outer leaflets[10]. The access to the furrow is controlled by the binding of $Ca^{2+}$ ions to a proximal site[10,15,16] that is situated within the inner third of the lipid bilayer and is constituted mainly by five conserved acidic residues located on three adjacent transmembrane helices ($\alpha6$-8)[10,17]. As revealed in structures obtained by cryo-electron microscopy (cryo-EM), the distinction between TMEM16 channels and scramblases is manifested in a conformational difference of $\alpha$-helices forming the subunit cavity[11]. The helix $\alpha4$, lining one edge of the open subunit cavity in the lipid scramblase nhTMEM16, has rearranged in TMEM16A to come in contact with $\alpha6$ on the opposite edge to form an aqueous pore that is for a large part shielded from the membrane. This ion conduction pore has an hourglass shape with wide aqueous cavities leading into a narrow neck from both sides of the membrane[12]. Anions are presumably conducted through the narrow neck with most of their coordinating water stripped, a process that is compensated for electrostatically by positively charged residues located at the extra- and intracellular entry of the neck[11].

Both pores in the dimeric protein act independently with respect to activation and ion conduction[18,19]. Activation of each TMEM16A pore appears to be controlled by two distinct mechanisms that are both mediated by the same $Ca^{2+}$ binding event. In the absence of $Ca^{2+}$, the repulsion between negatively charged residues in the vacant binding site causes the rearrangement of $\alpha6$ thereby facilitating the access of $Ca^{2+}$ from the intracellular side[12]. Binding of $Ca^{2+}$ to this vacant site initiates activation by reverting the negative electrostatic potential at the inner entrance, thereby lowering the barrier for anions during conduction[20]. At the same time, the bound $Ca^{2+}$ ions offer interactions with residues on $\alpha6$, causing its rearrangement around a glycine hinge. This is followed by presumed additional conformational changes that lead to the opening of a steric gate that was proposed to be located within the narrow neck[12].

Evidence for the location of the gate was provided from studies showing that the intracellular pore entrance retains its accessibility to small MTS reagents in the closed conformation in the absence of $Ca^{2+}$, whereas the neck remains inaccessible to the same reagents even in the activated state of the channel[12]. Despite the described evidence of a gate in the constricted pore region, the exact location of residues that obstruct ion flow in the closed conformation and their detailed spatial rearrangements during activation have remained elusive. This ambiguity is partly a consequence of the subtle conformational differences at this site between the $Ca^{2+}$-bound and -free structures of TMEM16A and the fact that the former might not display a fully conductive state. To clarify these open questions and define the residues involved in activation, we have engaged in a comprehensive characterization of point mutants by patch-clamp electrophysiology supported by structural studies. Our study reveals the location of a hydrophobic gate at the intracellular entry to the neck that controls ion conduction in TMEM16A thereby contributing to the tight regulation of its open probability.

## Results

**Comprehensive mutational analysis of the narrow pore region.** We have previously confined the location of a physical gate, which obstructs the ion conduction path in the closed conformation of TMEM16A, to the narrow neck region of the pore[12]. To identify residues forming this gate, we performed systematic mutagenesis of amino acids situated on helices enclosing the constricted region above the intracellular vestibule and of selected positions surrounding the $Ca^{2+}$ binding site (Fig. 1, and Supplementary Figs. 1 and 2, Supplementary Tables 1–3). We reasoned that residues contributing to a gate would face the pore and that truncation of their sidechains would increase the relative stability of conducting compared to non-conducting conformations of the channel. Such stabilization of an open state (or destabilization of a closed state) should be reflected in a change of the $Ca^{2+}$ potency, which in a ligand-gated channel is dependent on both the initial $Ca^{2+}$ binding step and subsequent coupled conformational changes[21]. Since most of the investigated positions are located remote from the $Ca^{2+}$ binding site and are thus unlikely to substantially interfere with $Ca^{2+}$ binding to the closed state, a left-shift of the $Ca^{2+}$ concentration-response relationship would reflect the relative stabilization of an open pore conformation by a mutation, which in severe cases would be accompanied by detectable basal activity. Conversely, a right-shift towards higher $Ca^{2+}$ concentrations would indicate a relative stabilization of the closed conformation and an unaltered $Ca^{2+}$ concentration-response relationship would correspond to no change in the distribution of states for a given mutation.

Our study has located strongly right-shifting mutants in the vicinity of the $Ca^{2+}$ binding site, certain moving parts of $\alpha6$ and interacting regions between $\alpha$-helices 5, 6, 7, and 8 (Fig. 1b–e). In addition, clusters of residues with a moderate rightward shift surround the pore at the extracellular part of the neck at the border to the outer vestibule (Fig. 1b, c). In contrast, residues whose mutation to alanine increases the $Ca^{2+}$ potency are lining the pore at the lower part of the neck region at the boundary to the intracellular vestibule (Fig. 1c, d). With G644P and Q649A, we have previously identified two mutations on $\alpha6$ with activating phenotype[12,20]. Both amino acids are located either at or close to a hinge region on $\alpha6$ that permits large conformational changes of this helix upon $Ca^{2+}$ binding (Fig. 1d). In our analysis, we now find additional mutations that stabilize the open state surrounding the intracellular opening of the narrow neck (Fig. 1d). At the inner pore, a cluster of hydrophobic residues, formed by Ile 550, Ile 551, both located on $\alpha4$, and Ile 641, located on $\alpha6$ and facing the pore on the opposite side, strongly determines the stability of the closed state as their alanine mutants show the most dramatic left-shifts in the $EC_{50}$ that are accompanied by the appearance of pronounced basal activity (Figs. 1d and 2a, b). Branching off from Ile 641 lies a zone of secondary residues, formed by Phe 589, Tyr 593, and Phe 712, that also help stabilize the closed state, whose alanine mutants exhibit considerable but somewhat less pronounced leftward shifts in the $EC_{50}$ and minimal basal activity (Figs. 1d and 2a, c).

We next investigated the current-voltage relationships of basal currents, which reflect the distribution of energy barriers along the permeation path. The strongly outwardly rectifying basal currents observed in the mutants I550A, I551A, and I641A (Fig. 2d, e) in

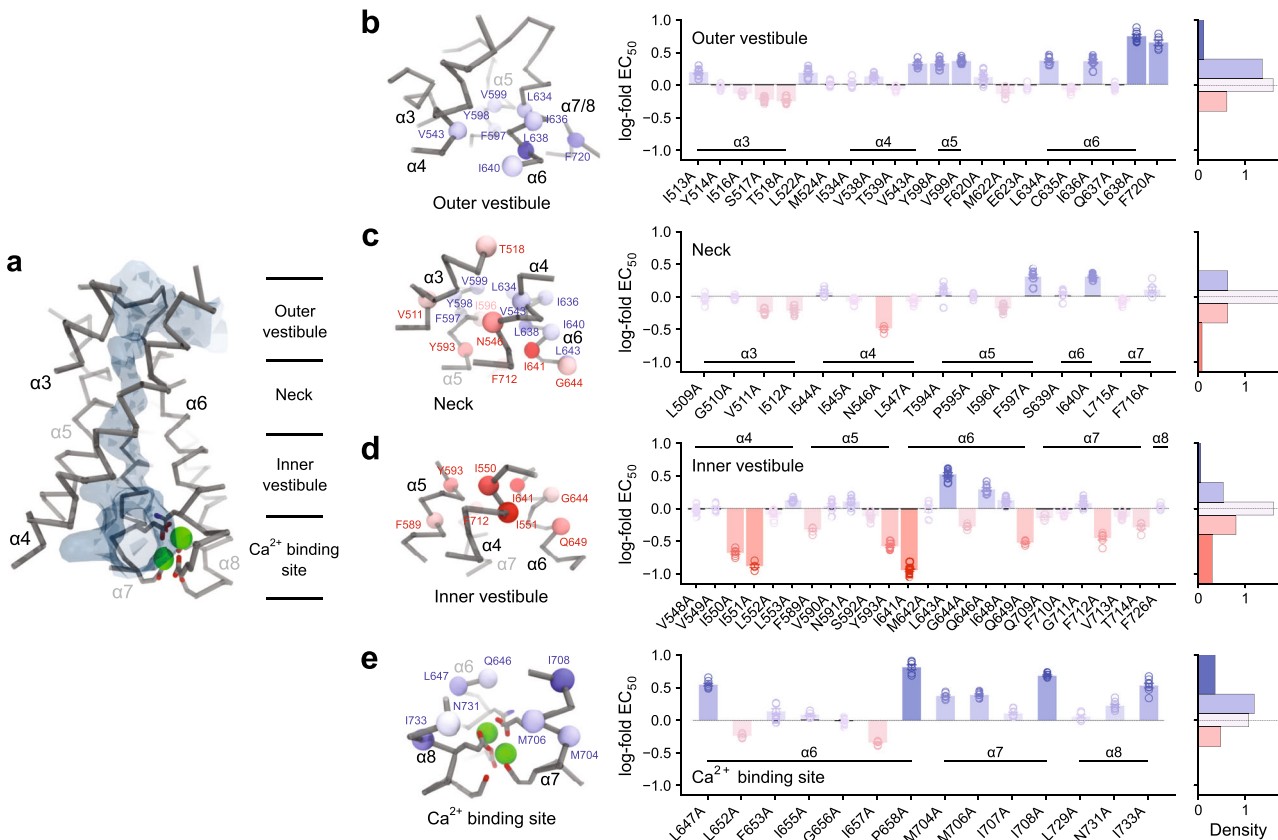

**Fig. 1 Characterization of pore residues by systematic mutagenesis. a** $C\alpha$ representation of the pore contained in a single subunit of TMEM16A (PDB: 5OYB) with different regions indicated. Blue surface encloses the water-accessible volume of the pore calculated in HOLE[53] with a probe radius of 1.15 Å. **b**–**e** Summary of $Ca^{2+}$ concentration-response relationships of Ala mutants in different regions of the pore. **b** Outer vestibule, **c** neck, **d** inner vestibule, and **e** $Ca^{2+}$ binding site. Red indicates a left-shift, and blue a right-shift in the $EC_{50}$. Left, sections of the pore with $C\alpha$ atoms of selected mutated residues shown as spheres and colored according to the effect on $Ca^{2+}$ potency. Center, $Ca^{2+}$ potencies of mutants. The logarithm of the fold-change in $EC_{50}$ of each investigated residue compared to wild type (WT) is shown. Individual measurements are displayed as circles, bars show averages of the indicated number of patches shown in Supplementary Tables 1–3, and errors are SEM. Right, histogram of $EC_{50}$ shifts in the corresponding region. **a**, **e** $Ca^{2+}$-binding residues are shown as sticks and bound $Ca^{2+}$ ions as green spheres.

the absence of $Ca^{2+}$ resemble the corresponding behavior in the mutants G644P and Q649A and most likely originate from the large repulsive energy barrier at the intracellular entry of the neck, which hampers ion conduction in the open pore of the apo state that we described previously[20]. This electrostatic barrier acts in addition to a physical gate to prevent ion conduction in the wild-type channel in the absence of $Ca^{2+}$ (refs. [12,20]). In the $Ca^{2+}$-bound state, a slight inward rectification in mutants I641A and I550A and a moderate outward rectification in mutants I551A and the previously identified Q649A corroborates the location of these residues on the anion permeation path in the open pore (Fig. 2d, e). Together, our results have revealed the distinct functional clusters around the narrow neck region of TMEM16A involved in channel activation. Whereas residues stabilizing the open state are placed in the upper part of the neck and around the $Ca^{2+}$ binding site (Fig. 1b, e), residues forming a gate that stabilizes the closed pore conformation, including three isoleucines (Ile 550, Ile 551, and Ile 641), which show the strongest energetic contribution (i.e., the most pronounced left shifts in the concentration-response relation and the appearance of basal activity), are located between the inner part of the neck and the intracellular vestibule (Figs. 1d and 2a). The observed effect of mutating these isoleucines is consistent with the presence of bulky hydrophobic residues at the intracellular entrance to the neck functioning as steric and hydrophobic barriers that prevent ion conduction in the closed state of the channel. While a moderate

widening of this region upon $Ca^{2+}$ binding was already found in cryo-EM structures of the protein[12], the functional data presented here imply a possible further expansion of the pore to be fully conductive.

**Cryo-EM structures of wild-type TMEM16A and an activating mutant**. To characterize the structural relationship between residues constituting the gate and address how their mutation to alanine stabilizes the open state, we studied WT and the mutant I551A by cryo-EM (Supplementary Figs. 3–5, Table 1). We and others have previously determined the structure of a $Ca^{2+}$-bound conformation of TMEM16A[12,13]. However, since the protein was purified in the continuous presence of $Ca^{2+}$ and in absence of the lipid $PI(4,5)P_2$, both of which promote the transition into a non-conducting conformation in patch-clamp experiments[18,22–26], it was uncertain whether these structures would exhibit features of such a non-conducting state. We thus collected cryo-EM data for wild-type TMEM16A, which was purified in the absence of divalent cations, pre-incubated with a water-soluble $PI(4,5)P_2$ analog and where $Ca^{2+}$ was added briefly before sample vitrification. The structure determined at 3.7 Å is virtually indistinguishable from the earlier TMEM16A structure in the $Ca^{2+}$-bound state, suggesting that the previously applied conditions did not affect the observed conformation of the protein (Supplementary Fig. 6a). Despite the presence of diC8-$PI(4,5)P_2$ in the sample, no densities could be

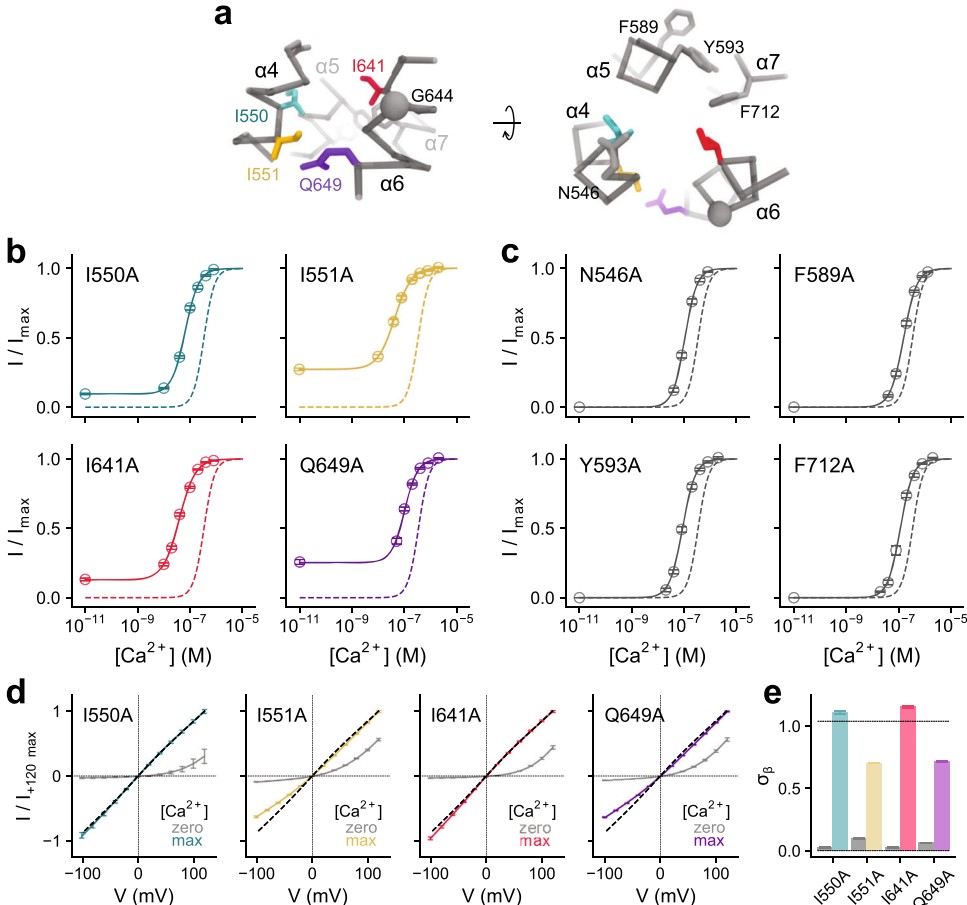

**Fig. 2 Functional properties of mutants forming the gate. a** Cα representation of the entrance to the narrow region of the pore in TMEM16A. Sidechains of selected residues are displayed. The relationship of views is indicated. **b, c** Concentration-response relations of selected mutants of the inner neck region with left-shifted $EC_{50}$ for **b**, residues showing basal activity and **c**, residues not showing pronounced basal activity. Data are averages of the indicated number of patches shown in Supplementary Tables 1–3, errors are SEM. Solid lines are fits to the Hill equation. Dashed lines show the relation of WT. **d** Instantaneous I-V relations of mutants that display basal activity at zero and saturating $Ca^{2+}$ concentrations. Dashed lines show the relation of WT at saturating $Ca^{2+}$ concentrations. Data are averages of 5, 10, 7, and 13 patches for I550A, I551A, I641A, and Q649A respectively, errors are SEM. Solid lines are fits to a model of ion permeation (Eq. 2). **e** Values of $\sigma_\beta$, corresponding to the relative rate of conduction at the inner pore close to the $Ca^{2+}$ binding site (see "Methods"), for mutants displaying basal activity at zero and saturating $Ca^{2+}$ concentrations. Dashed line indicates the value of WT at saturating $Ca^{2+}$ concentrations. Bars indicate the best-fit values of the averaged data shown in **d**. Errors are 95% confidence intervals.

confidently attributed to the lipid analogue, which could either reflect a weakening of $PI(4,5)P_2$ binding to the channel in a deter-gent environment or, alternatively, be a consequence of its intrinsic mobility which impedes its identification at the observed resolution of the data. Since the purified protein conducts anions after lipo-some reconstitution[12] and undergoes structural rearrangements that are characteristic of activation within seconds of exposure to $Ca^{2+}$, the vitrified protein likely displays a conformation that is func-tionally relevant. Still, since at its constriction the diameter of the pore is narrower than the size of permeant anions, its full opening might have been precluded in a detergent environment. Due to the potentially incomplete pore opening, the observation that α6 adopts an activated conformation suggests that the protein might display features of a pre-open intermediate (i.e., a $Ca^{2+}$-activated non-conducting state) that we describe in an accompanying manu-script[27], although we cannot exclude a closer resemblance to an inactivated state that is adopted upon dissociation of $PI(4,5)P_2$. We next investigated the structure of the constitutively active mutant I551A in the presence of $Ca^{2+}$ (Fig. 3a). Although at a lower resolution of 4.1 Å (Supplementary Fig. 4), the general correspon-dence of the mutant structure to WT emphasizes their equivalent properties in the $Ca^{2+}$-bound state (Supplementary Fig. 6a).

Finally, we determined the structure of the mutant I551A in the absence of $Ca^{2+}$ to elucidate the structural features underlying its activating behavior (Supplementary Fig. 5). The structure at 3.3 Å provides a detailed view of a constitutively active mutant in a ligand-free state (Fig. 3a). Its overall conformation generally resembles the structure observed for the $Ca^{2+}$-free state of WT[12] except for a pronounced conformational difference at the intracellular half of α6 (Fig. 3b, c and Supplementary Fig. 6a). As a consequence of the electrostatic repulsion between negatively charged residues in the vacant binding site, the helix has changed its conformation compared to the $Ca^{2+}$-bound state, although in a different direction and to a lesser extent than observed for the ligand-free WT (Fig. 3b, c). Compared to the $Ca^{2+}$-free wild-type structure where the intracellular part of α6 has moved towards α4, in I551A it moved outwards by about 30° in a direction away from α4 (Fig. 3c), resembling a conformation that was also found for the $Ca^{2+}$-free state of the lipid scramblase TMEM16F[14]. As for WT, the π-helical region of α6 located below the gating hinge Gly 644 found in the $Ca^{2+}$-bound state has also relaxed towards a canonical α-helix in I551A (Fig. 3d), despite the difference in the conformation of α6 between the apo structures (Fig. 3c and Supplementary Fig. 6a). The loss of density beyond Asn 651 in

**Table 1 Cryo-EM data collection, processing, refinement, and validation statistics.**

| | TMEM16A WT Ca²⁺ +diC8-PI(4,5)P₂ | TMEM16A I551A apo +diC8-PI(4,5)P₂ | TMEM16A I551A Ca²⁺ +diC8-PI(4,5)P₂ |
|---|---|---|---|
| **Data collection and processing** | | | |
| Magnification | 49,407 | 49,407 | 49,407 |
| Voltage (kV) | 200 | 200 | 200 |
| Electron dose (e⁻/Å²) | 53 | 53 | 53 |
| Defocus range (μm) | −0.5 to −2.0 | −0.5 to −2.0 | −0.5 to −2.0 |
| Pixel size (Å) | 1.012 | 1.012 | 1.012 |
| Symmetry | C2 | C2 | C2 |
| Initial particle images | 1,214,923 | 462,927 | 166,511 |
| Final particle images | 23,887 | 138,320 | 34,234 |
| Map resolution (Å) | 3.7 | 3.3 | 4.1 |
| FSC threshold | 0.143 | 0.143 | 0.143 |
| Map local resolution range (Å) | 5.5–3.4 | 5.5–3.1 | 6.9–4.0 |
| **Refinement** | | | |
| Initial model used | PDB ID: 5OYB | PDB ID: 5OYG | PDB ID: 5OYB |
| Model resolution (Å) FSC$_{model}$ = 0.5 | 3.8 | 3.4 | 4.2 |
| Model resolution range (Å) | 80–3.8 | 80–3.4 | 80–4.2 |
| Map sharpening B factor (Å²) | −34 | −76 | −86 |
| **Model composition** | | | |
| Nonhydrogen atoms | 11,770 | 11,462 | 11,764 |
| Protein residues | 1436 | 1402 | 1436 |
| Ligands | 4 | 0 | 4 |
| **B factors (Å²)** | | | |
| Protein | 51.9 | 27.3 | 55.2 |
| Ligand | 25.7 | | 40.9 |
| **r.m.s. deviations** | | | |
| Bond lengths (Å) | 0.005 | 0.005 | 0.006 |
| Bond angles (°) | 0.85 | 0.83 | 0.93 |
| **Validation** | | | |
| MolProbity score | 1.8 | 1.7 | 2.0 |
| Clash score | 7.4 | 4.7 | 9.2 |
| Poor rotamers (%) | 0.8 | 1.0 | 0.3 |
| **Ramachandran plot** | | | |
| Favored (%) | 94.1 | 92.8 | 92.0 |
| Allowed (%) | 5.9 | 7.2 | 8.0 |
| Disallowed (%) | 0 | 0 | 0 |

I551A presumably reflects the increased flexibility of the helix in the absence of Ca²⁺ (Supplementary Fig. 6b, c).

The observed differences between the Ca²⁺-free structures of WT and I551A emphasize the functional interaction between the gate region and the intracellular half of α6 in TMEM16A. In WT, the electrostatic repulsion between acidic residues at the vacant Ca²⁺ binding site located on α7 and α8 and Glu 654 on α6 in part underlies the conformational change of α6 in the Ca²⁺-free state. The relative stabilization of the open state in the mutant enables α6 to adopt a seemingly more activated conformation (with partially straightened helix α6), in part overcoming the electrostatic repulsion at the vacant Ca²⁺ binding site (Fig. 3b, c, e). Thus, it is conceivable that the observed structure displays features relevant to the mutant's basal activity where, even in the absence of Ca²⁺, the movement of α6 couples to the narrow neck to stabilize a conductive state of the channel.

**Systematic mutational analysis of residues in the gate region.** Since the truncation of sidechains of three juxtaposed isoleucine residues at the intracellular pore narrowing exerted a strong influence on the opening of the channel, we decided to investigate the collective properties of mutations of the triplet on channel activation. Although Ile 550 and Ile 551 both residing on α4 and Ile 641 on α6 are not in direct physical contact in the apo state[12], they frame the opposite sides of the narrow pore (Fig. 2a) and we thus anticipate a potential cooperative interaction between the

three residues in controlling anion access to the neck region. Initially, we probed the systematic variation of the hydrophobic volume of these sidechains by successively mutating the respective isoleucines to either valine, thereby removing a single methyl group, or alanine, which removes three methyl groups at once without changing the aliphatic character of the residue (Supplementary Fig. 7a). For all mutants, we investigated concentration-response relationships to determine the potency of Ca²⁺, which we relate to the distribution of states. In these experiments, the stability of the open state shows an inverse correlation with the number of methyl groups within the isoleucine triad (Fig. 4a, b, Supplementary Table 4) that encloses the pore, further supporting the role of these residues as being part of a hydrophobic gate that excludes water and ions in the closed conformation. As predicted from gating schemes that are based on allosteric transitions (see Supplementary Note), the EC$_{50}$ first decreases with decreasing number of methyl groups but eventually saturates and reaches a limiting EC$_{50}$ that defines the highest binding affinity for the agonist Ca²⁺ (Fig. 4b). Although the mutants of the three residues show EC$_{50}$ shifts to varying degree, with mutations of Ile 641 generally exerting the strongest effect, this trend can be described with a simple Monod-Wyman-Changeux (MWC) model[28] assuming that the mutations affect only the gating step (Fig. 4b, Supplementary Note, "Methods"). From this analysis, we obtained that on average each methyl group contributes 0.83 ± 0.21 kcal/mol in stabilizing the closed state, which coincides with the range expected for van der Waals interactions[29].

The hydrophobic nature of the gate is also reflected in constructs where the respective isoleucines are replaced with amino acids with stronger polar character (Supplementary Fig. 7b). Irrespective of the size of the introduced residues, the mutations cause an increase in the potency of Ca²⁺ that depends on the hydrophilicity of the replacement (Fig. 4c, d, Supplementary Table 5). The stability of the open state correlates with increasingly more favorable hydration energy within the isoleucine triad for sidechains that coarsely retain steric volume (Fig. 4c, d), again consistent with the formation of a hydrophobic gate that excludes water and ions in the closed conformation. From an analysis similar to the one used for their truncation and under the assumption that the energetic effect of mutations is proportional to the hydration energy of sidechains[30], the fractional contribution (i.e., the proportionality constant) of substituted residues in stabilizing the open state was estimated to be 0.37 ± 0.11.

Finally, we analyzed our data with respect to interactions between the investigated residues, which are evident from non-additive shifts in the EC$_{50}$ amongst double and triple mutants within the supposed log-linear range (Fig. 4b, Supplementary Fig. 7c). Although our previous analysis has assumed additivity of energetic effects to account for the general trend, we find deviations that indicate a functional coupling between the gate residues, which we quantified in a series of double-mutant cycles (Fig. 5a). No pronounced coupling was observed between the adjacent residues Ile 550 and Ile 551, as I551V retains much of its effect in stabilizing the open state when introduced on an I550V background (Fig. 5b). This translates into a near-zero coupling energy ($G_{coupling}$) between the two residues, indicating that both residues act independently in stabilizing the closed state (Fig. 5c). In contrast, the introduction of I550V or I551V individually on an I641V background renders these mutations less effective in further stabilizing the open state (Fig. 5b). Consequently, the coupling energy significantly deviates from zero (Fig. 5c), which suggests functional interactions between Ile 641 and either Ile 550 or Ile 551 in stabilizing the closed state. Triadic coupling within the gate region becomes apparent when a third mutation is introduced, which is manifested in the non-zero difference

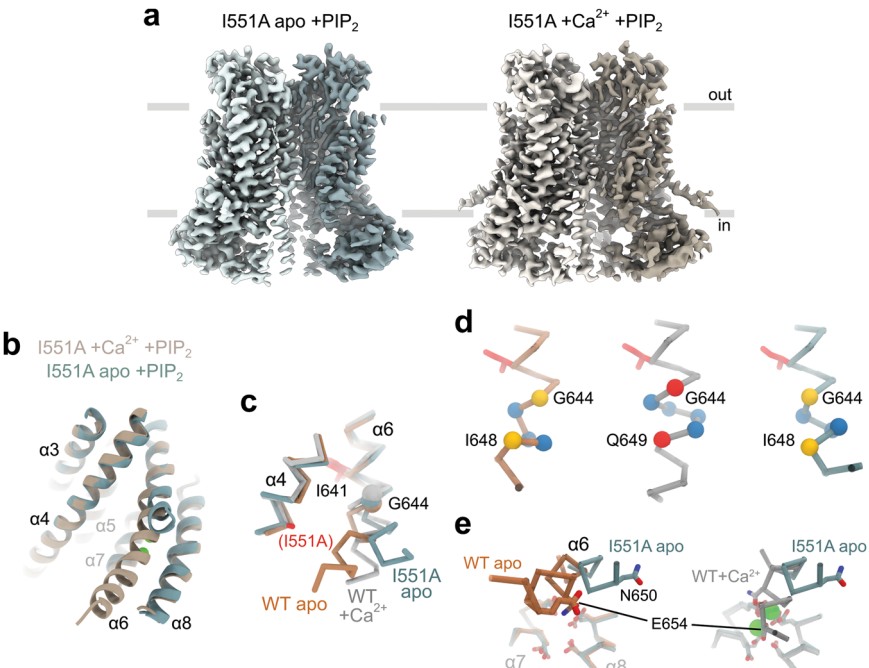

**Fig. 3 Structural features of a constitutively active mutant. a** Cryo-EM map of mouse TMEM16A-I551A in the absence (left) and presence (right) of 1 mM $Ca^{2+}$ supplemented with 1.5 mM diC8-PI(4,5)$P_2$ in GDN at 3.3 and 4.1 Å respectively. The view is from within the membrane, with the extracellular side at the top. **b** Superposition of the pore region (α3–α8) of the apo and $Ca^{2+}$-bound mutant structures in ribbon representation. The view is rotated by ~45° around the dimer axis compared to **a**. **c** Superposition of α4 and α6 of the indicated structures in Cα representation. The Cα of Gly 644 is shown as sphere, the sidechains of Ile 641 and the mutation I551A in the mutant structure as sticks. The apo and $Ca^{2+}$-bound structures of WT were previously reported[12] (PDB: 5OYG and 5OYB respectively). **d** Section of α6 around Gly 644. Yellow spheres depict respective pairs of hydrogen-bonded positions in α-helix conformation, red spheres depict a pair of interacting residues in π-helix conformation, and blue spheres indicate the Cα positions in between. **e** Superposition of the $Ca^{2+}$ binding sites of indicated structures viewed from within the membrane. The protein is shown in Cα representation, and sidechains of $Ca^{2+}$ binding residues as sticks. **d**, **e** The coloring of the Cα-traces is as in **c**. **b**, **e** $Ca^{2+}$ ions in the $Ca^{2+}$-bound structure are shown as green spheres.

between coupling energies ($\Delta G_{coupling}$) of mutant pairs in a triple-mutant cycle (Fig. 5d). Collectively, our functional characterization thus defines the importance of hydrophobic interactions within the isoleucine triad at the intracellular entrance to the narrow neck region in controlling gating in TMEM16A (Fig. 5e).

**Rearrangements of the gate in the open state**. To gain further insight into the role of the gating residues during ion conduction through the open channel, we investigated the impact of mutations of Ile 550, Ile 551, and Ile 641 on current-voltage relationships in the $Ca^{2+}$-bound state. Residues at the constriction facing the pore are expected to interfere with conduction when extra sidechain volume is introduced, leading to current rectification due to elevated energy barriers at the site of mutation. Moreover, as the nature of rectification depends on the position of rate-limiting barriers[11], this analysis provides further evidence for the location of the gate with respect to the anion conduction path (Fig. 6a). Enlarging the sidechain volume of Ile 641 on α6 by mutation to Met and Phe increases local energy barriers at the intracellular pore entrance and the neck, as manifested in the pronounced outward rectification of currents (Fig. 6b), indicating that this residue remains oriented towards the pore in the open conformation. The gradual effect of Ile 641 mutations on conduction suggests incremental hindrance of permeation that depends on the size of the sidechain (Fig. 6b, c). In contrast, equivalent mutations of Ile 550 and Ile 551, which are located on the opposing helix α4, do not lead to strong rectification (Fig. 6d), suggesting that, unlike Ile 641, these residues do not contribute to rate-limiting energy barriers for conduction in the open state in a size-dependent manner (Fig. 6e). Instead, they might have

retracted further from the pore constriction than observed in the $Ca^{2+}$-bound conformation of TMEM16A, corroborating with a plausibly more extended rearrangement of the pore in a conducting state. The distinct effects of titrating sidechain volume of residues on α4 and α6 are thus consistent with non-equivalent spatial relationships between the gating residues in the open and closed states of the pore.

## Discussion

In the present study, we were interested in the location of the gate in TMEM16A that prevents ion conduction in the closed state of the channel. To this end, we have investigated the effect of mutations of residues lining the narrow part of the pore and have identified several positions where a mutation to alanine stabilizes the open state of the channel, the majority of which cluster at the intracellular part of the neck region. The strongest effect was observed for three hydrophobic residues, two of which occupy neighboring positions on α4 (Ile 550 and Ile 551) at the border to the wider intracellular vestibule and another located on α6 (Ile 641) on the opposite side of the pore slightly further up in the extracellular direction (Figs. 1 and 2). Sidechain truncation of any of the three residues by mutation to alanine causes a strong increase in the potency of $Ca^{2+}$ and results in pronounced basal activity.

Although in the known structures, the sidechains of the three residues controlling anion access to the narrow neck region appear not to be in van der Waals contact, mutant cycle analysis suggests a functional coupling between them. Such coupling could proceed by an indirect mechanism via residues around the gate region, as alanine mutations of residues in the vicinity of Ile

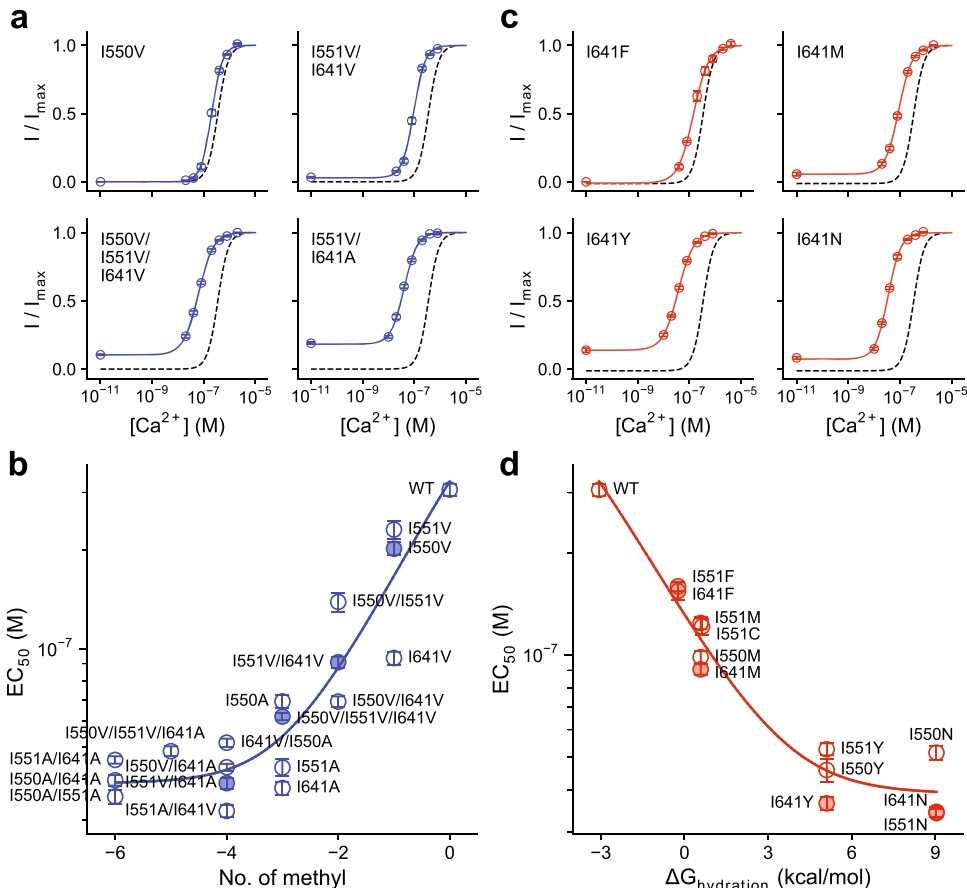

**Fig. 4 Energetic contribution of hydrophobic volume and hydration energy of gate residues. a** Selected concentration-response relations of mutants with decreasing hydrophobic volume of gate residues. **b** Relationship between $EC_{50}$ change and hydrophobic volume decrease. The effective contribution of each methyl group was estimated to be $0.83 \pm 0.21$ kcal/mol in stabilizing the closed state. **c** Selected concentration-response relations of mutants with increasing hydrophilicity of gate residues. **d** Relationship between $EC_{50}$ change and hydration energy. The fractional contribution of the residues' hydration energy was estimated to be $0.37 \pm 0.11$ in stabilizing the open state. **a**, **c** Data are averages of the indicated number of patches shown in Supplementary Tables 4 and 5 respectively, errors are SEM. Solid lines are fits to the Hill equation. Dashed lines are the relation of WT. **b**, **d** Filled symbols correspond to the mean $EC_{50}$ of the mutants shown in **a** and **c** respectively. Data are averages of the indicated number of patches shown in Supplementary Tables 4 and 5 respectively, errors are SEM. Solid line is a fit to an MWC-type gating model (Eqs. 4–8, see "Methods"). The two series were fitted globally with shared binding constants. The errors of the estimates correspond to 95% confidence intervals.

641 such as Phe 712 and its immediate interacting partners Ile 596 and Tyr 593 all result in a similar but somewhat smaller stabilization of the open pore (Figs. 1 and 2). Alternatively, the coupling between sidechains that are not in direct contact could also be mediated via the surrounding solvent. Solvent-mediated coupling is consistent with the effect of mutants either decreasing the hydrophobic volume or increasing the hydrophilicity of the respective sidechains, which both result in a destabilization of the closed state (Fig. 4). In the open state, the hydrophobic interactions that exclude the access of water to the gate region break down leading to the opening of a water-accessible path. As a result, the relative roles of the three residues on the ion permeation path have changed as illustrated by the distinct effects of increasing sidechain volume on conduction, where mutations of Ile 641 but not of Ile 550 and Ile 551 severely perturb current-voltage relationships (Fig. 6). This is consistent with a widening of the pore at the intracellular entry to the neck upon channel opening (Fig. 7a), and a redistribution of Ile 550 and Ile 551 to establish an interaction network that is described in further detail in an accompanying study[27]. Both features are evident in the structures of the $Ca^{2+}$-free and $Ca^{2+}$-bound states, although these structures might not display the full range of conformational changes leading to pore opening.

The chemical nature of the gate in TMEM16A is notable in light of unrelated channel architectures. The presence of bulky hydrophobic residues at the pore constriction, which form a physical barrier for ion permeation, is a recurrent theme in ion channels and has been identified to close the pore in diverse families such as $K^+$ channels, pentameric ligand-gated ion channels, and bestrophins to name a few[31–33]. Although frequently found in van der Waals distance, a direct contact between the respective residues is not mandatory, since once their respective location narrows the pore diameter below a certain threshold, spontaneous dewetting of the region can result, which imposes a further energetic penalty for ion permeation[34,35]. A similar mechanism might also control gating in TMEM16A to restrict conduction in the closed states and to contribute to the low open probability of WT in the absence of $Ca^{2+}$ (Fig. 7a).

Besides the mutual relationship between residues of the gate region described above, the cryo-EM structures of the mutant I551A also revealed determinants related to the coupling of the gate to the $Ca^{2+}$ binding site. The described interaction between the two regions is manifested in the conformation $\alpha$6 in I551A, which likely underlies the observed basal activity. Whereas the mutant resembles WT in the $Ca^{2+}$-bound state, the $Ca^{2+}$-free structure of I551A exhibits pronounced differences compared to

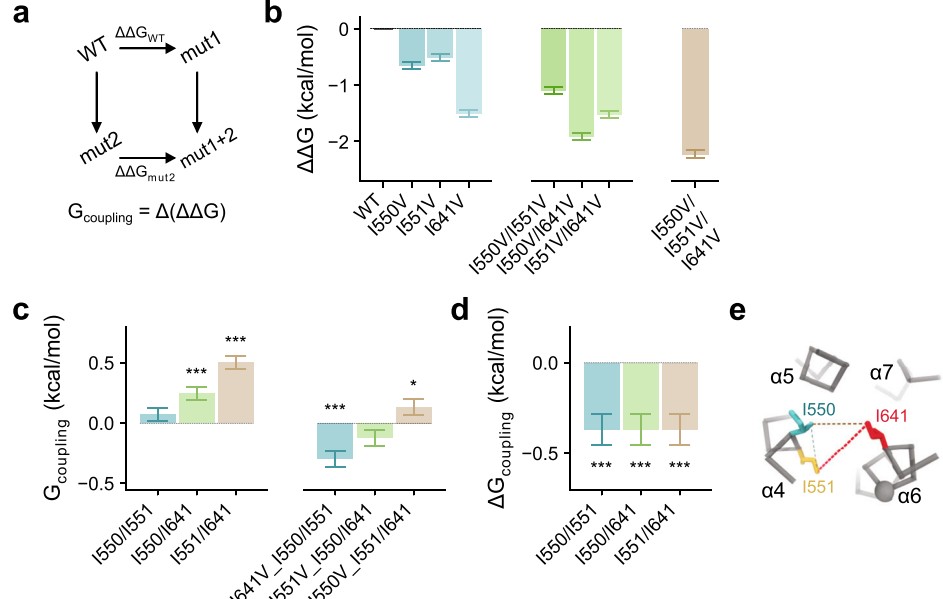

**Fig. 5 Functional coupling within the triadic gate. a** Schematic illustration of mutant cycle analysis. **b** $\Delta\Delta G$ of the displayed mutants calculated by fitting their concentration-response relations to an MWC-type gating model (Eqs. 4–6, 9, and 11–12). Bars indicate $\Delta\Delta G$ calculated from the best-fit values of the averaged data shown in Supplementary Fig. 7c. Errors correspond to 95% confidence intervals. **c** Coupling energy ($G_{coupling}$ or $\Delta\Delta G$) measured in double-mutant cycles in the background of WT (left) or indicated mutants (right). Bars indicate the values calculated from the best-fit values shown in **b** using Eq. 13. **d** Change in coupling energy ($\Delta G_{coupling}$ or $\Delta\Delta\Delta G$) between the cycles displayed in **c**. Bars indicate the values calculated from the values shown in **c** using Eq. 14. **e** Cα representation of the inner pore entrance viewed from the extracellular side. Dashed lines depict functional coupling between the displayed residues with a thickness approximately corresponding to the respective coupling energies shown in **c**, left. **c**, **d** Errors are standard errors. Asterisks indicate significant deviation from zero in a two-sided one-sample t-test (from left to right, **c** ***$p = 2e-5$; ***$p = 2e-16$; ***$p = 2e-5$; *$p = 0.043$; **d** ***$p = 3e-5$ for each value).

the corresponding structure of WT. As for WT, the electrostatic repulsion between negatively charged residues in the vacant binding site causes a dissociation of the intracellular half of α6 from its tight interaction with α8 observed in the $Ca^{2+}$-bound structure, leading to a rearrangement of the helix and increased mobility (Fig. 3b, c). However, in I551A this movement is less pronounced than in WT and it proceeds in a different direction. As a result, α6 remains in a position that is closer to its fully activated conformation, thereby lowering the energetic penalty for channel opening. Despite this difference in its position, the previously described relaxation of α6 from a π- to an α-helix upon dissociation of $Ca^{2+}$ and the consequent loss of an interaction with α8 (ref. [12]) are both observed in the mutant structure, further emphasizing the conformational strain due to π-helix formation in the $Ca^{2+}$-bound state that is surmounted by interactions with the bound agonist. Importantly, these observations suggest that pore opening can proceed without the transition of α6 into a straightened π-helix conformation (Fig. 7b). Nevertheless, a moderate decrease in the open probability in the apo state compared to the $Ca^{2+}$-bound state of the mutant (see accompanying manuscript[27]) suggests that the difference in α6 conformation might affect gating, and we thus cannot exclude some impact of the mutation on the conformation of the open pore of the apo protein.

In summary, our study has identified a gate region that stabilizes the closed pore of TMEM16A and provided evidence for its interaction with the $Ca^{2+}$ binding element of α6 (Fig. 7). In the closed state, the proximity of hydrophobic residues at the intracellular entrance of the narrow neck prevents access of water and ions to this location. Upon activation, the breakdown of hydrophobic interactions in the gate region in conjunction with further conformational rearrangements of the protein lead to the expansion of the constricting neck region in the anion-conducting state. The structures presented here likely delineate a general mechanism for activation where the sequential rearrangement of the intracellular half of α6 couples to the narrow neck region of the pore to open the gate although they likely do not show the full extent of activation. The described steric mechanism acts in conjunction with the previously described electrostatic gate[20] to ensure a tight control of TMEM16A activity in response to cellular signaling events. A related mechanism might underlie activation in lipid scramblases of the TMEM16 family[14–16], where coupling of α6 upon $Ca^{2+}$ binding is transmitted to α4 leading to the dissociation of both helices from each other and the opening of a membrane-accessible hydrophilic furrow, which catalyzes the shuffling of lipid headgroups across the membrane.

## Methods

**Molecular biology and cell culture.** HEK293T cells (ATCC CRL-1573) were maintained in Dulbecco's modified Eagle's medium (DMEM; Sigma-Aldrich) supplemented with 10 U ml⁻¹ penicillin, 0.1 mg ml⁻¹ streptomycin (Sigma-Aldrich), 2 mM L-glutamine (Sigma-Aldrich), and 10% FBS (Sigma-Aldrich) in a humidified atmosphere containing 5% $CO_2$ at 37 °C. HEK293S GnTI⁻ cells (ATCC CRL-3022) were maintained in HyClone HyCell TransFx-H medium (GE Healthcare) supplemented with 10 U ml⁻¹ penicillin, 0.1 mg ml⁻¹ streptomycin, 4 mM L-glutamine, 0.15% poloxamer 188 (Sigma-Aldrich), and 1% FBS in an atmosphere containing 5% $CO_2$ at 190 rpm at 37 °C. Mutations were introduced using a modified QuikChange method[36] and were verified by sequencing. Primers are listed in Supplementary Table 6.

**Protein expression and purification.** For the preparation of protein used in cryo-EM experiments, GnTI⁻ cells were transiently transfected with wild-type mouse TMEM16A or the point mutant TMEM16A-I551A complexed with Poly-ethylenimine MAX 40 K (formed in non-supplemented DMEM medium at a w/w ratio of 1:2.5 for 30 min). Immediately after transfection, the culture was supplemented with 3.5 mM valproic acid. Cells were collected 48-h post-transfection, washed with PBS, and stored at −80 °C until further use. Protein purification was carried out at 4 °C and was completed within 12 h. The protein was purified in $Ca^{2+}$-free buffers and was supplemented with 1 mM free $Ca^{2+}$ when indicated

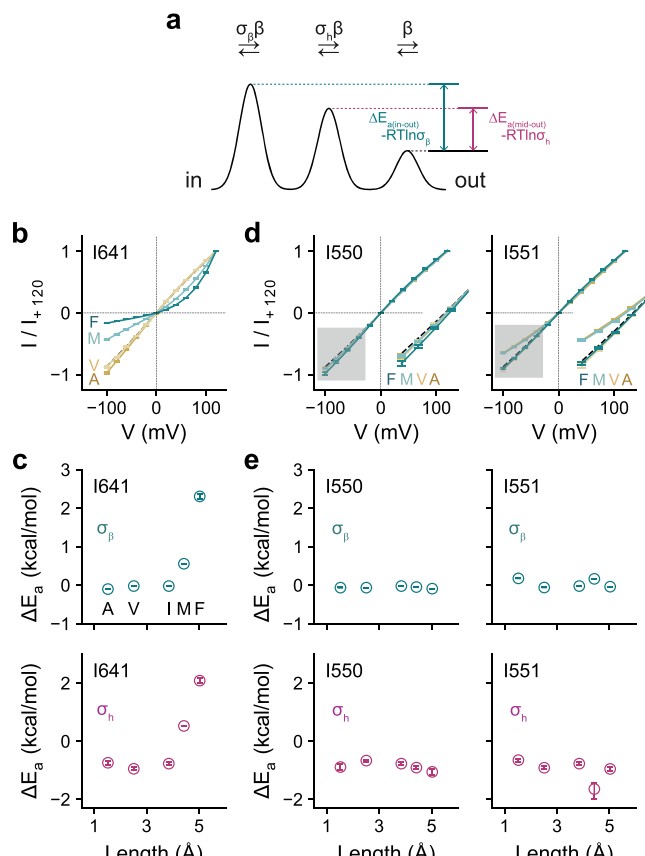

**Fig. 6 Effect of sidechain volume on ion conduction in the open state.**
**a** Energy profile of a minimal ion permeation model to account for the I-V relations of TMEM16A. **b** Instantaneous I-V relations of mutations of Ile 641 with increasing sidechain volume at saturating $Ca^{2+}$ concentrations.
**c** Energy barrier relative to the outermost barrier in the conduction path at the inner pore entrance (top) and at the middle of the pore (bottom) for Ile 641. **d** Instantaneous I-V relations of mutations of Ile 550 and Ile 551 with increasing sidechain volume at saturating $Ca^{2+}$ concentrations. Inset shows a magnified view of the shaded region. **e** Energy barrier relative to the outermost barrier in the conduction path at the inner pore entrance (top) and at the middle of the pore (bottom) for the residues Ile 550 and Ile 551.
**b**, **d** Data are averages of 7, 6, 9, and 7 patches (I641), 6, 7, 5, and 11 patches (I550), and 8, 10, 7, and 10 patches (I551) for A, V, M, and F respectively, errors are SEM. Solid lines are fits to a model of ion permeation (Eq. 2). Dashed lines show the relation of WT. **c**, **e** Data are calculated using Eq. 3 from the best-fit values of the averaged data shown in **b** and **d** respectively, errors are 95% confidence intervals.

during cryo-EM sample preparation. Cells were resuspended and solubilized in 150 mM NaCl, 5 mM EGTA, 20 mM HEPES, 1× cOmplete protease inhibitors (Roche), 40 μg ml⁻¹ DNase (AppliChem), 2% GDN (Anatrace) at pH 7.4 by gentle mixing for 2 h. The solubilized fraction was obtained by centrifugation at 16,000 × g for 30 min. After filtration with 0.5 μm filters (Sartorius), the supernatant was incubated with streptavidin UltraLink resin (Pierce, Thermo Fisher Scientific) for 2 h under gentle agitation. The beads were loaded onto a gravity column and were washed with 60 column volume of SEC buffer containing 150 mM NaCl, 2 mM EGTA, 20 mM HEPES, 0.01% GDN at pH 7.4. The bound protein was eluted by incubating the beads with 3 column volume of SEC buffer supplemented with 0.25 mg ml⁻¹ 3C protease for 30 min. The eluate was concentrated using a 100 kDa cutoff filter, filtered through a 0.22 μm filter, and loaded onto a Superose 6 10/300 GL column (GE Healthcare) pre-equilibrated with SEC buffer. Peak fractions containing the protein were pooled, concentrated, filtered through a 0.22 μm filter, and used immediately for cryo-EM sample preparation.

**Electron microscopy sample preparation and data collection**. 2.5 μl of purified protein, concentrated to ~2 mg ml⁻¹ and pre-incubated with 1.5 mM diC8-PI(4,5) P₂ (Echelon Biosciences) for at least 30 min at 4 °C, was applied onto holey carbon

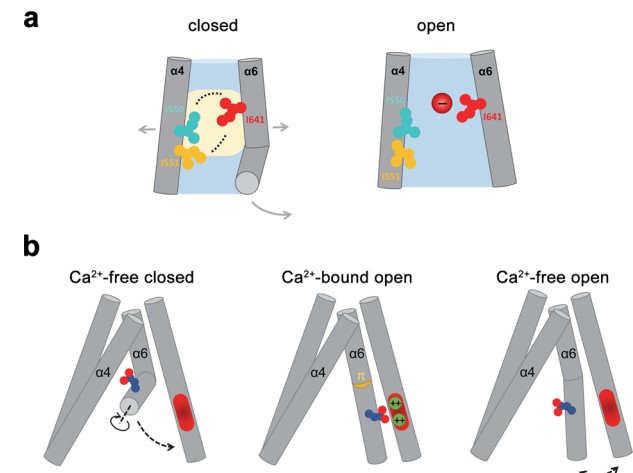

**Fig. 7 Relationship between non-conducting and conducting conformations. a** Schematic illustration of the hydrophobic gate at the inner entrance of the narrow neck that prevents ion conduction in the closed state (left). Functional interactions between hydrophobic residues are indicated by dashed lines. Beige area indicates putative de-wetted region that excludes water in the closed conformation. In the open conformation (right), the residues of the gate have dissociated leading to a widening of the pore and a retraction of gate residues on α4. **b** Relationship between conducting and non-conducting conformations in the presence and absence of $Ca^{2+}$. In the non-conducting apo conformation of WT (left), the intracellular half of α6 has moved away from the $Ca^{2+}$ binding site. Upon $Ca^{2+}$ binding, α6 rearranges its conformation by moving towards the $Ca^{2+}$ binding site. The subsequent rotation around the helix axis, to bring a residue in contact with bound $Ca^{2+}$ ions, introduces a strained π-helix conformation. The movement of α6 couples to the gate region to open the channel (center). The coupling between the gate and α6 is illustrated in the structure of a gate mutant showing basal activity in the absence of $Ca^{2+}$ (right). In this case, α6 has approached the vacant binding site, opening the gate without transiting to a strained π-helix conformation.

grids (Quantifoil Au R1.2/1.3, 300 mesh). Immediately prior to sample application, grids were glow-discharged at 15 mA for 30 s. After sample application, grids were blotted for 3–5 s with a blot force setting of 0 at 4 °C at 100% humidity, plunge-frozen in a liquid propane/ethane mixture using a TFS Vitrobot Mark IV (Thermo Fisher Scientific), and stored in liquid nitrogen until further use. For samples with $Ca^{2+}$, SEC buffer supplemented with 18 mM $CaCl_2$ was mixed with the concentrated purified protein at a ratio of 1:5 (resulting in a final free $Ca^{2+}$ concentration of 1 mM) immediately before sample application and plunge-freezing.

Data collection was performed on a Talos Arctica (Thermo Fisher Scientific) operated at 200 kV, and equipped with a BioQuantum energy filter (20 eV slit width) and a K2 direct electron detector (Gatan). EPU2 (Thermo Fisher Scientific) was used for automated data collection at a calibrated pixel size of 1.012 Å/pixel and a nominal defocus range of –0.8 to –1.9 μm. Each movie contained 60 frames with an exposure time of 9 s and a total dose of 53 e⁻/Å² (0.883 e⁻/Å²/frame). Holes were selected based on a Digital Micrograph script determining ice thickness at the grid square level[37] (manuscript in preparation). Data were on-the-fly analyzed using FOCUS[38] and targeting parameters were adjusted if necessary.

**Image processing**. For all collected movies, beam-induced motion correction was done by MotionCor2 (ref. [39]) and the CTF was determined on aligned movie stacks using CTFFind4 (ref. [40]). Both processes were run through FOCUS[38] which was further used to curate the data set based on CTF resolution estimation (<4 Å), defocus value (–0.5 μm to –2.0 μm), ice thickness (20–50 nm), and general appearance.

For the $Ca^{2+}$-free mutant I551A, 3979 movies were collected from which 3606 movies were selected for further processing. crYOLO[41] was used for automated particle picking resulting in an initial set of 462,927 particles. All following steps were executed in Relion 3.0 (ref. [42]). Particles were extracted with a box size of 210 pixels and binned 3x (70 pixels, 3.036 Å/pixel). After several rounds of 2D classification where CTFs until the first peak were ignored, 286,846 particles were selected and re-extracted with a box size of 256 pixels and binned 2x (128 pixels, 2.024 Å/pixel). The re-extracted particles were subjected to a 3D classification without symmetry applied and ignoring the CTFs until the first peak using a

cryoSPARC[43] map low-pass filtered to 40 Å as the initial reference. One of the five classes containing 160,844 particles represented the structure of the channel. The particles were re-extracted unbinned and refined against the respective class low-pass filtered to 40 Å resulting in a 3.6 Å map. If not otherwise stated, all refinements were done without symmetry applied and were continued after convergence with a mask excluding the detergent micelle. After CTF refinement, which did not result in an improvement of the resolution, the particles were further classified in 3D without realignment using the previous refined map low-pass filtered to 20 Å. The main class of three containing 152,049 particles refined to 3.5 Å. After Bayesian polishing, which resulted in a slight improvement of the resolution, the particles were subjected to another round of CTF refinement followed by a 2D classification to remove residual bad particles. The final 138,320 particles were refined with C2 symmetry applied and resulted in a 3.3 Å map.

In the case of the Ca$^{2+}$-bound mutant I551A, 624 movies were collected of which 599 movies were selected. 166,511 particles picked by crYOLO were extracted unbinned with a box size of 256 pixel and subjected to several rounds of 2D classification in cryoSPARC. The cleaned stack of 56,578 particles was used for the generation of three initial models in cryoSPARC. 34,234 particles yielding the best model of the channel were used for a homogeneous refinement in cryoSAPRC using C2 symmetry, which resulted in a 4.6 Å reconstruction. All following steps were performed in Relion 3.0. C2 symmetry was applied throughout refinement and a mask excluding the detergent micelle was introduced after initial convergence. The best particles from cryoSPARC were re-extracted and refined against the respective map low-pass filtered to 40 Å resulting in a 4.1 Å map. The followed CTF refinement and Bayesian polishing did not change the overall resolution, but the b-factor applied during postprocessing was improved from −150 to −86 Å$^2$.

For the Ca$^{2+}$-bound wild-type structure, 2189 movies were collected of which 1764 were selected. Particles were picked using template-free Laplacian-of-Gaussian-based auto-picking in Relion 3.0 and were extracted with a box size of 256 pixels and binned 2x (128 pixels, 2.024 Å/pixel). After several rounds of 2D classification with and without ignored CTFs until the first peak, 100,190 particles were selected and subjected to a 3D classification without symmetry applied using an initial model generated in Relion 3.0 that was low-pass filtered to 50 Å. One of the eight classes containing 34,477 particles represented the structure of the channel, which was subjected to another round of 3D classification without symmetry applied. The resulting 25,361 particles were re-extracted unbinned and refined against the respective class low-pass filtered to 50 Å with C2 symmetry applied, resulting in a 4.1 Å map. The particle images were subjected to both CTF refinement and Bayesian polishing, which resulted in a slight improvement of the resolution to 4.0 Å. The particles were further classified in 3D once without realignment using the refined map low-pass filtered to 50 Å. The final 23,887 particles were refined with C2 symmetry applied and resulted in a 3.7 Å map.

All resolution estimations followed the gold standard of two independently refined half maps[44] and applying the 0.143 FSC cut-off[45]. Local resolutions were determined with Relion's own implementation. Directional FSCs were calculated using the 3DFSC server[46].

**Model building and refinement.** Initial models were obtained by docking the corresponding wild-type TMEM16A structures (PDB: 5OYG and 5OYB respectively) into the densities of the apo and Ca$^{2+}$-bound TMEM16A-I551A using Chimera. The models were iteratively rebuilt in Coot[47] and refined in Phenix[48]. The geometry of the final models was evaluated using MolProbity[49]. For model validation, the FSCs between the refined model and the final map and/or the summed half-maps were determined (FSC$_{model}$ and FSC$_{sum}$ respectively) and a threshold of 0.5 was used[45]. To monitor potential over-fitting, random shifts up to 0.5 Å were introduced to the coordinates of the final model, followed by refinement in Phenix against the first unfiltered half-map. The FSC between this shaken-refined model and the first half-map (FSC$_{work}$) was compared with that against the second half-map (FSC$_{free}$), which was not used in the refinement. Figures were prepared using Chimera[50], ChimeraX[51], and VMD[52].

**Electrophysiology.** HEK293T cells were transfected with 3 μg DNA per 6 cm Petri dish using the calcium phosphate co-precipitation method, and were used within 24–96 h after transfection. Recordings were performed on inside-out patches excised from HEK293T cells expressing the construct of interest. Patch pipettes were pulled from borosilicate glass capillaries (O.D. 1.5 mm, I.D. 0.86 mm, Sutter Instrument) and were fire-polished with a microforge (Narishige) before use. Pipette resistance was typically 3–8 MΩ when filled with the recording solutions detailed below. Seal resistance was typically 4 GΩ or higher. Voltage-clamp recordings were made using Axopatch 200B, Digidata 1550, and Clampex 10.6 (Molecular devices). Analog signals were filtered with the in-built 4-pole Bessel filter at 10 kHz and were digitized at 20 kHz. Solution exchange was achieved using a gravity-fed system through a theta glass pipette mounted on an ultra-fast piezo-driven stepper (Siskiyou). Liquid junction potential was found to be consistently negligible given the ionic composition of the solutions and was therefore not corrected. All recordings were performed at 20 °C.

A symmetrical ionic condition was used throughout. Stock solution with Ca$^{2+}$-EGTA contained 150 mM NaCl, 5.99 mM Ca(OH)$_2$, 5 mM EGTA, and 10 mM HEPES at pH 7.40. Stock solution with EGTA contained 150 mM NaCl, 5 mM

EGTA, and 10 mM HEPES at pH 7.40. Free Ca$^{2+}$ concentrations were adjusted by mixing the stock solutions at the required ratios calculated using the WEBMAXC program (http://web.stanford.edu/~cpatton/webmaxcS.htm). Patch pipettes were filled with the stock solution with Ca$^{2+}$-EGTA, which has a free Ca$^{2+}$ concentration of 1 mM.

**Estimating EC$_{50}$.** Concentration-response relations were fitted to the Hill equation,

$$I/I_{max} = \frac{1}{1 + \left(\frac{EC_{50}}{[Ca^{2+}]}\right)^h} \quad (1)$$

where $I/I_{max}$ is the normalized current response, EC$_{50}$ defines the concentration at which activation is at its half-maximum, and $h$ is the Hill coefficient.

**Analysis of current-voltage (I-V) relations.** I-V data were fitted to a minimal permeation model that accounts for the fundamental biophysical behavior of mTMEM16A as described previously[11],

$$I = zFAe^{\frac{zFV}{2nRT}} \frac{c_i - c_o e^{-\frac{zFV}{RT}}}{e^{-zFV\frac{n-1}{nRT}} + \left(\frac{1}{\sigma_h}\right)\frac{1 - e^{-zFV\frac{n-2}{nRT}}}{e^{\frac{zFV}{nRT}} - 1} + \frac{1}{\sigma_\beta}} \quad (2)$$

where $I$ is the current, $n$ is the number of barriers, $c_i$ and $c_o$ are the intracellular and extracellular concentrations of the charge carrier, $z$ is the valence of Cl$^-$, $V$ is the membrane voltage, and $R$, $T$, and $F$ have their usual thermodynamic meanings. $A = \beta_0 \nu$ is a proportionality factor where $\beta_0$ is the value of $\beta$ when $V = 0$ and $\nu$ is a proportionality coefficient that has a dimension of volume. $\sigma_h$ and $\sigma_\beta$ are respectively the rate of barrier crossing at the middle and the innermost barriers relative to that at the outermost barrier ($\beta$). The best-fit values of $\sigma_\beta$ and $\sigma_h$ at zero and saturating Ca$^{2+}$ concentrations were used to calculate $\Delta E_{a(\sigma\beta)}$ and $\Delta E_{a(\sigma h)}$, the difference between the activation energy at the innermost barrier and the middle barrier relative to that of the outermost respectively, using

$$\Delta E_{a(\sigma_\beta)} = -RT \ln \sigma_\beta$$
$$\Delta E_{a(\sigma_h)} = -RT \ln \sigma_h \quad (3)$$

**Mechanism and parameter estimation.** To describe the effect of sidechain properties and to characterize functional interactions between residues forming the inner gate, the energetic differences governing the potency shifts were obtained. For that purpose, we fitted the concentration-response relations to a minimal activation model consisting of a closed and an open state with two identical binding steps.

$$
\begin{array}{ccc}
& L_0 & \\
C_0 & \leftrightarrow & O_0 \\
K_{d(C)} \updownarrow & & \updownarrow K_{d(O)} \\
C_1 & \leftrightarrow & O_1 \\
K_{d(C)} \updownarrow & & \updownarrow K_{d(O)} \\
C_2 & \leftrightarrow & O_2 \\
& L_2 &
\end{array}
$$

A feature of this model is that the three levels of conductance/current $(i,j,k)$ associated with the degree of Ca$^{2+}$ occupancy $(0,1,2)$ can be incorporated. The normalized current response is given by

$$I/I_{max} = \frac{iP_{O_0} + jP_{O_1} + kP_{O_2}}{kP_{O_{x\to\infty}}} \quad (4)$$

where

$$P_{O_0} = \frac{L_0}{Q_C + L_0 Q_O}$$

$$P_{O_1} = \frac{L_0 \frac{x}{K_{d(O)}}}{Q_C + L_0 Q_O}$$

$$P_{O_2} = \frac{L_0 \left(\frac{x}{K_{d(O)}}\right)^2}{Q_C + L_0 Q_O}$$

$$P_{O_{x\to\infty}} = \frac{L_2}{1 + L_2} \quad (5)$$

$$Q_C = 1 + \frac{x}{K_{d(C)}} + \left(\frac{x}{K_{d(C)}}\right)^2$$

$$Q_O = 1 + \frac{x}{K_{d(O)}} + \left(\frac{x}{K_{d(O)}}\right)^2$$

$x$ is the ligand concentration, $L_0$ is the forward equilibrium constant between the closed and open states at zero Ca$^{2+}$ occupancy, and $K_d$ is the dissociation equilibrium constant with the subscripts $C$ and $O$ denoting the closed and open

states respectively. $P$ denotes the occupancy of the indicated state, and $L_2$ is the forward equilibrium constant between the closed and open states at maximum $Ca^{2+}$ occupancy. The gating constant at zero $Ca^{2+}$ occupancy ($L_0$) was obtained from microscopic reversibility

$$L_0 = L_2 \left( K_{d(O)} \right)^2 / \left( K_{d(C)} \right)^2 \tag{6}$$

where $L_2$ for WT ($L_{2\text{WT}}$) was determined from $P_{O_{x \to \infty}}$ estimated from non-stationary noise analysis (see accompanying manuscript[27]). Because of normalization, the current levels $i$ and $j$ can be expressed as a fraction of $k$. The values for $i/k$ were determined using the ratios of the current at +80 mV obtained from the instantaneous I-V plots at zero and saturating $Ca^{2+}$ concentrations. $\Delta E_{a(\sigma\beta)}$ and $\Delta E_{a(\sigma h)}$ values linearly interpolated from those obtained for apo and double occupancy were used to calculate the I-V plot expected for single occupancy, which was used to estimate the value of $j/k$.

We estimated the energetic contributions of the residues forming the gate from experiments where their chemical properties were titrated. For effects originating from hydrophobic volume, we assumed that the gating constant $L_2$ can be expressed as a function of the number of methyl groups ($n_{\text{Me}}$) and that on average each methyl group has an identical effective energetic contribution ($\Delta G_{\text{Me}}$)

$$L_2(\Delta n_{\text{Me}}) = L_{2\text{WT}} e^{-\Delta n_{\text{Me}} \Delta G_{\text{Me}}/RT}$$
$$\Delta n_{\text{Me}} = n_{\text{Me(mut)}} - n_{\text{Me(WT)}} \tag{7}$$

where the subscript mut denotes mutant. For hydration effects, we assumed that a fraction ($\delta$) of the residues' hydration energy contributes towards the gating equilibrium

$$L_2(\Delta\Delta G_{\text{hydration}}) = L_{2\text{WT}} e^{\delta\Delta\Delta G_{\text{hydration}}/RT}$$
$$\Delta\Delta G_{\text{hydration}} = \Delta G_{\text{hydration(mut)}} - \Delta G_{\text{hydration(WT)}} \tag{8}$$

The values of hydration energy were taken from Kyte and Doolittle[30]. To obtain the chemical parameters ($\Delta G_{\text{Me}}, \delta$), the sum of squares between the logarithm of $EC_{50}$ values computed numerically from the independent variables of the experiments ($\Delta n_{\text{Me}}, \Delta\Delta G_{\text{hydration}}$) and the logarithm of experimental $EC_{50}$ values was minimized. The effects of titrating the number of methyl groups and hydration energy were optimized globally to obtain a unique set of binding constants ($K_{d(O)}$, $K_{d(c)}$) that can describe both datasets. A more detailed description of the model is provided as Supplementary Note.

For mutant-specific effects, changes in the gating constant $L_2$ were incorporated as

$$L_{2\text{mut}} = L_{2\text{WT}} e^{-\Delta G_{\text{mut}}/RT} \tag{9}$$

Because the same set of binding constants ($K_{d(O)}, K_{d(c)}$) was sufficient to account for the effect of the mutants, these were used as shared parameters, resulting in one free parameter per mutant ($\Delta G_{\text{mut}}$). For parameter estimation, a series of concentration-response relations corresponding to the individual $\Delta G_{\text{mut}}$ was computed. The sum of squares between each of these relations and their experimental counterparts was calculated, and the total sum of squares was minimized. A more thorough examination of the errors associated with this analysis is presented as Supplementary Note.

The variance of the best-fit parameters was obtained from the diagonal elements of the variance-covariance matrix

$$\mathbf{H}^{-1} = \left( \mathbf{J}^T \cdot \mathbf{J} \right)^{-1} \tag{10}$$

multiplied by

$$\frac{\sum(\text{residual})^2}{n_{\text{data}} - n_{\text{parameter}}}$$

where $\mathbf{H}$ and $\mathbf{J}$ are the Hessian and Jacobian matrices at the least squares estimates respectively, the superscript $T$ indicates transpose, and $n$ are the number of data points and parameters respectively. The square root of the variance was used to approximate the standard deviation error, from which the 95% confidence interval was computed.

**Triple-mutant cycle analysis.** The free energy of transition ($\Delta G$) was calculated from the forward equilibrium constant using

$$\Delta G = -RT \ln L \tag{11}$$

where $R$ and $T$ have their usual thermodynamic meanings and $L$ is the forward equilibrium constant. The change in the free energy of transition ($\Delta\Delta G$) caused by a mutation was calculated as

$$\Delta\Delta G^{(0-X,Y)} = \Delta G^{(0,Y)} - \Delta G^{(X,Y)}$$
$$\Delta\Delta G^{(X,0-Y)} = \Delta G^{(X,0)} - \Delta G^{(X,Y)}$$
$$\Delta\Delta G^{(0-X,0)} = \Delta G^{(0,0)} - \Delta G^{(X,0)}$$
$$\Delta\Delta G^{(0,0-Y)} = \Delta G^{(0,0)} - \Delta G^{(0,Y)} \tag{12}$$

where $X$ and $Y$ indicate the two residues of interest and 0 denotes a mutation. The redundant energetic contribution between $X$ and $Y$ or coupling energy ($G_{\text{coupling}}$ or

$\Delta\Delta\Delta G^{\text{XY}}$) was calculated using either the $X$ or $Y$ mutations

$$\Delta\Delta\Delta G^{\text{XY}} = \Delta\Delta G^{(0-X,0)} - \Delta\Delta G^{(0-X,Y)}$$
$$= \left( \Delta G^{(0,0)} - \Delta G^{(X,0)} \right) - \left( \Delta G^{(0,Y)} - \Delta G^{(X,Y)} \right) \tag{13}$$

The dependency on a third residue ($\Delta G_{\text{coupling}}$ or $\Delta\Delta\Delta\Delta G^{\text{XYZ}}$) was quantified as the difference between $\Delta\Delta\Delta G^{\text{XY}}$ and $\Delta\Delta\Delta G^{\text{XY}}$ in the presence of an additional mutation ($\Delta\Delta\Delta G^{\text{XY}}_{Z \to 0}$) using

$$\Delta\Delta\Delta\Delta G^{\text{XYZ}} = \Delta\Delta\Delta G^{\text{XY}}_{Z \to 0} - \Delta\Delta\Delta G^{\text{XY}} \tag{14}$$

For brevity, the superscripts are dropped throughout the text. The standard error ($\sigma$) of the parameter estimates for each subtraction was propagated as described in the Data analysis and statistics section. Deviation of $\Delta\Delta\Delta G^{\text{XY}}$ or $\Delta\Delta\Delta\Delta G^{\text{XYZ}}$ from zero was detected using a two-sided one-sample t-test with a significance level of 0.05.

**Data analysis and statistics.** Electrophysiology data were extracted and organized using Clampfit 10.6 (Molecular Devices) and Excel (Microsoft). Experimental $EC_{50}$ values were obtained using Prism 8 (GraphPad). Model analysis and numerical calculations were performed using NumPy (https://numpy.org) and SciPy (https://scipy.org). Parameter optimization was performed using the described sum of squares objective functions with the least_squares function in SciPy, which also computes the Jacobian matrix that was used to estimate the 95% confidence intervals. Experimental data consisting of individual measurements are presented as mean ± SEM. Estimated parameters are presented as best-fit ± 95% confidence interval unless otherwise stated. Standard error uncertainties of estimated parameters were propagated using

$$\sigma_{(a+b \text{ or } a-b)} = \sqrt{\sigma_a^2 + \sigma_b^2}$$
$$\frac{\sigma_{(ab \text{ or } a/b)}}{|f(a,b)|} = \sqrt{\left(\frac{\sigma_a}{|a|}\right)^2 + \left(\frac{\sigma_b}{|b|}\right)^2} \tag{15}$$

The one-sample t-test, with a significance level of 0.05, was used to analyze deviation from zero. Statistical analysis was performed using either Prism 8 and/or NumPy/SciPy.

**Reporting summary.** Further information on research design is available in the Nature Research Reporting Summary linked to this article.

## Data availability

Data supporting the findings of this manuscript are available from the corresponding authors upon reasonable request. A reporting summary for this Article is available as a Supplementary Information file. Source data are provided with this paper. Maps, half-maps, and masks have been deposited in the EMDB and can be found under the entries. https://www.ebi.ac.uk/pdbe/entry/emdb/EMD-12025 (WT-Ca²⁺), https://www.ebi.ac.uk/pdbe/entry/emdb/EMD-12026 (I551A-apo), https://www.ebi.ac.uk/pdbe/entry/emdb/EMD-12027 (I551A-Ca²⁺). The respective atomic models are available in the PDB under PDB 7B5C (WT-Ca²⁺), PDB 7B5D (I551A-apo), PDB 7B5E (I551A-Ca²⁺).

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

## Acknowledgements

We thank all members of the Dutzler and Paulino lab for help at various stages of the project. This work was supported by a grant of the European Research Council (ERC no. 339116, AnoBest) to Raimund Dutzler, a grant of the Dutch Research Council (NWO Veni grant 722.017.001 and the NWO Start-Up grant 740.018.016) to C.P., and a Forschungskredit of the University of Zurich (grant no FK-18-048) to A.K.M.L.

## Author contributions

A.K.M.L. conceived the study, performed molecular biology and electrophysiology experiments, analyzed electrophysiology data, and prepared protein and cryo-EM samples. J.R. collected cryo-EM data. J.R. and A.K.M.L. jointly processed cryo-EM data, constructed and refined models. C.P. overlooked cryo-EM data collection. A.K.M.L. and R.D. prepared an initial draft of the manuscript and all coauthors contributed to the final manuscript.

## Competing interests

The authors declare no competing interests.
