## [Peer Review File · Nature Communications]

Reviewer #1 (Remarks to the Author):

Gating the pore of the calcium-activated chloride channel TMEM16A

The calcium-activated chloride channel TMEM16A is an important channel from a physiological perspective. The channel protein is distinct from other families of channels both in terms of its three dimensional structure and in terms of its mechanisms. In this nice study, the authors address the molecular mechanisms of “gating”, by which the ion pore permits chloride conduction when a gate is open and prevents ion permeation when a gate is closed. The authors used scanning mutagenesis to identify amino acids that affect the equilibrium between the open and closed conformations and identify three hydrophobic residues (Ile 550, Ile 551, Ile 641) in particular that when mutated to alanine bias a conductive (open) form of the channel. These three amino acids contribute to the walls of the putative ion conduction pore. Structural studies, including cryo-EM structures of one of the mutants in both the calcium-bound and calcium-free conformations, provide insights into the calcium-dependent activation mechanism and the effects that calcium binding has on the pore of the channel. The authors propose that these three amino acids function as major constituents of the gate of the channel. The studies are well executed and technically sound. The studies provide new insights into the molecular mechanisms of gating in this family of channels. The combination of structural biology with rigorous electrophysiological approaches provides a rare depth of mechanistic insight into ion channel function.

The following specific comments are meant to help the authors improve their manuscript.

I think it would be helpful for the authors to more clearly define what they mean by “gating” and what is meant by a “gate”. This comment relates to several specific comments below. Some of these comments are clarified by the accompanying manuscript, but it would make this manuscript stronger to address them here.

The authors note, in lines 60-62, “Despite the described evidence of a gate in the constricted pore region, the exact location of residues that obstruct ion flow in the closed conformation and the detailed spatial rearrangements during activation have remained elusive.” In the structures of Ca-bound and Ca-free TMEM16A, determined by the authors in this study and previous ones, the amino acids identified (I550, I551, I641) do not seem to undergo conformational changes and the dimensions of the regions of the pore that they surround do not seem to change much either. The authors may wish to clarify to the reader why their data supports the conclusion that this region operates as the “gate” if it doesn’t seem to undergo conformational changes in these structures.

Line 111-113 – please provide reference for this statement.

Line 114. Is “outward rectification” the proper terminology for the small current seen at -100mV? The channels still seem to be inward rectifiers.

Line 120, phrase “which show the strongest energetic contribution” was a bit confusing/unclear. I assume you are referring to the greatest shift in the Ca-response curve. It would be helpful for the reader to clarify the connection that you are making between free energy differences and Ca-activation curves.

Minor point, on several occasions, the authors refer to mutations as stabilizing the open or closed states (e.g. line 97). It seems possible that a given mutation that biases the open state may do so by

either energetically stabilizing the open state or by destabilizing the closed state. Or vice versa for a mutation that biases a closed state.

Line 121-123, I found the phrase “latter observation” a bit unclear. Also, regarding that sentence: “The latter observation is consistent with the presence of bulky hydrophobic residues at the intracellular entrance to the neck functioning as steric and hydrophobic barriers that prevent ion conduction in the closed state of the channel.” I agree. However, the authors should discuss this hypothesis in the context of the observation that the conformations of the residues (I550, I551, I641) do not seem to change between the Ca-bound and Ca-free structures. If the dimensions of the pore at I550, I551, I641 do not change, how do the authors envision that they function as a gate? (This is the main question addressed in the accompanying manuscript, but it would help the reader to provide more context here.)

Line 182-185, “In these experiments, the stability of the open state shows an inverse correlation with the number of methyl groups within the isoleucine triad (Fig. 4a,b), further supporting the role of these residues as being part of a hydrophobic gate that excludes water and ions in the closed conformation.” The fact that mutations to the I550, I551, I641 residues biases the open state does not seem to necessarily indicate that these residues are part of the variable constriction in the channel that functions as a gate. Certainly their location relative to the putative ion conduction pore suggests that they may function in this regard, but it seems possible that their mutation favors an open conformation (e.g. through an allosteric mechanism) without necessarily indicating that the residues are part of the hydrophobic constriction that functions as a gate.

Line 191-193. A reference is needed.

Line 200-202. More information regarding what the authors mean by “fractional contribution of the hydration energy” would help readers not familiar with this terminology/methodology.

Line 227 – reference needed.

Lines 234-239: “In contrast, equivalent mutations of Ile 550 and Ile 551, which are located on the opposing helix $\alpha 4$, do not lead to strong rectification (Fig. 6d), suggesting that these residues do not contribute to rate-limiting energy barriers for conduction in the open state (Fig. 237 6e), and that they might instead have retracted from the pore.” The authors may wish to clarify what they mean by “retracted from the pore” since the conformations of Ile 550 and Ile 551 are very similar between the Ca-bound and Ca-free conformations of the channel. Perhaps they are hinting that the Ca-bound structure is not fully open?

Line 261-264, I did not understand the phrase “the gate dissociates”. (Again, this is clarified by the accompanying manuscript, but context would help here.)

Line 300: “In summary, our study has identified a gate region that stabilizes the closed pore of TMEM16A ...” I agree that the studies of the I550, I551, I641 residues suggest that this region stabilizes a closed (non-conductive) conformation of the channel. In line with the preceding comments, the authors may wish to clarify what they mean by “gate region” and how the gate operates (does it change in dimensions, for example, to allow or prevent ion conduction?). As this is the subject of their accompanying manuscript, they should at least mention this.

In Figures 1d and 6b, it would be helpful to show the wild type I-V curves as well.

The authors should comment on the absence/presence of density for PIP2 (diC8-PI(4,5)P2).

Reviewer #2 (Remarks to the Author):

The present manuscript investigates the gating mechanism of the TMEM16A calcium activated chloride channel. These channels are key players in human physiology as they regulate multiple processes, from smooth muscle excitability to epithelial salt reabsorption and fertilization. Despite intense structural and functional scrutiny, the molecular mechanisms underlying the opening and closing of these channels remains poorly understood, as all current structures report on the closed conformation of the channel. Past work suggested that ion permeation through the TMEM16A channel is regulated by the electrostatic profile of the pore and by the release of a hydrophobic constriction along the pore. Here, Lam and colleagues use functional mutagenesis and electrophysiology to probe the roles of pore-lining side chains and show that mutations at the hydrophobic gasket specifically facilitate channel activation by increasing the EC50 for Ca²⁺ and increasing the basal open probability in the absence of Ca²⁺. The structure of the I551A mutant in Ca²⁺-free conformation shows rearrangements around the α 6 helix that forms the Ca²⁺ binding site, while that of the mutant in the Ca²⁺-bound state is similar to that of the WT protein in the same conditions. Mutant cycle analysis shows that the residues forming the hydrophobic gasket are energetically coupled, and that the hydrophobicity of the side chains at these positions stabilizes the closed conformation of the pore.

Overall, this is an elegant and well-designed study that provides interesting insights into the opening mechanism of the TMEM16A channel. I have some requests for clarifications from the authors and would suggest they tone down some of their conclusions, that in some places seem to be a bit overstated relative to the data.

1) Did the authors account for the desensitization of the TMEM16A channel during their electrophysiological experiments? Do any of the mutants affect this process? Since the channel's pore appears to be in a non-conductive conformation in all structures (presented here and in previous work), how can the authors rule out the possibility that they all represent a desensitized conformation?

2) The authors present 3 new structures of TMEM16A: the WT channel in the presence of PIP2, and of the I551A mutant in Ca²⁺-bound and Ca²⁺-free states. Unfortunately, the insights that can be obtained from these 3 structures are limited: addition of PIP2 seems to do nothing, the I551A mutant has no effect in Ca²⁺ bound state and induces a small rearrangement of α 6 in the Ca²⁺ free state. The structures are over-interpreted.

- There is no evidence that the PIP2 structure represents an intermediate on the way to opening as the authors claim (line 138-139). There is no evidence that PIP2 binds to the detergent-solubilized channel protein, so that the protein could be in a desensitized state. Addition of PIP2 to the solution does not mean that it will bind to the protein, it is possible that PIP2 binding is weakened in the absence of a lipid membrane. To address this concern a structure of TMEM16A in a nanodisc supplemented with PIP2 would be needed. However, I realize that this might not be a straightforward or feasible experiment. Nonetheless, the authors should tone down their

statements on what the state represents and discuss the above-mentioned potential pitfall.

- The authors repeatedly refer to the I551A mutant as constitutively active (i.e. lines 140, 146, Fig. 3 legend). However, the authors data are not consistent with this claim: the I551A mutant is Ca²⁺ and voltage dependent. I think it is more appropriate to refer to this mutant as “activating”, as the authors do in other portions of the manuscript (i.e. in the abstract).

- I am not sure what mechanistic inferences can be drawn from the I551A mutant structures. The observation that the mutation affects the conformation of $\alpha 6$ is interesting and supports the notion this residue plays a role in gating. However, I am not sure how the authors can claim that “The relative stabilization of the open state in the mutant enables $\alpha 6$ to adopt a seemingly activated conformation...” (lines 164-165, a similar concept is posited in line 290). What supports the idea that $\alpha 6$ is in an activated conformation? Given that the pore is in a non-conductive conformation and that we do not know what the open TMEM16A pore looks like, this statement seems unwarranted.

The paragraph discussing the structure and the associated inferences should be drastically toned down.

- A minor concern is that the density of the $\alpha 6$ helix in the Ca²⁺-free I551A mutant map is quite weak. I wonder how confident are the authors that there is a π - to α -helix transition in this mutant.

- At what σ values are the densities shown in Extended Data Fig. 6?

3) The authors use mutant cycle analysis to show that the positions around the hydrophobic constriction are functionally coupled. They then use a Markov model to extract the energetic contributions of individual methyl groups in the pore to the EC50 for Ca²⁺ and to hydration energy of the pore. The use of this model is central to the present work, as it enables the authors to extract detailed, quantitative mechanistic information on the opening process. Overall, this analysis is very elegant and intriguing. However, the description of the Markov model in the main text is non-existent and is very succinct in the Methods section. As such, some of the non-obvious insights are cryptic (i.e. why does the EC50 first decrease with decreasing number of methyl groups and then saturates?) and it is very difficult to evaluate the robustness of the estimation of the parameters of the model. A thorough description of the model and of how parameters are estimated is needed. Further, the authors should document how robust the fitting process is (i.e. how coupled are the various parameters in the model? how unique is the choice of parameter set for each condition within the parameter space of the model?). I suggest the authors add an appendix to the manuscript where these critical issues are thoroughly discussed, and the model and its assumptions are validated. Without this documentation, I have concerns that the estimates of the modeled gating parameters might not be sufficiently precise and uniquely determined to enable the quantitative inferences on the energetic coupling and of the contributions of individual side chains to the gating process.

Reviewer #3 (Remarks to the Author):

Calcium-activated chloride channels are transmembrane proteins that play important roles in

various physiological processes, such as neuronal signaling and smooth muscle contraction. TMEM16A encodes one such channel, whose activity is up-regulated upon Ca²⁺ binding, with different responding outcomes depending on the types of tissues where TMEM16A is expressed. Unraveling the detailed mechanism of channel activation and regulation will greatly improve our understanding of this type of proteins, and benefit development of therapeutic strategies targeting TMEM16A, given its involvement in certain diseases, such as cystic fibrosis.

Structures of TMEM16A in its apo-form and Ca²⁺-bound form have been determined by cryo-electron microscopy in previous studies, revealing conformational alteration between the two states. In addition, a gating regulatory mechanism for the ion conductance was suggested. As a follow-up study, in this manuscript, Lam et al. carried out sophisticated electrophysiology experiments and extensive data analysis for systematic mutated variants of TMEM16A. The authors pinpoint amino acid residues that cause the stabilization of the closed conformation and the open conformation, respectively. Interestingly, a series of hydrophobic residues (I550, I551, and I641) were found to have a significant effect on stabilizing TMEM16A in its closed conformation. The authors attributed the effect of double or triple mutation of these three residues to their functional coupling. Moreover, cryo-EM structures of the I551A variant of TMEM16A were determined in the absence or presence of Ca²⁺, respectively, with the Ca²⁺-free I551A mutant showing similar conformation but noticeable difference. The experiments in this study were well executed and data analysis was performed carefully. The manuscript was very well written and easy to follow, and it was a delight to read the manuscript. The findings by the authors further our understanding of the regulatory mechanism of TMEM16A to a deeper extent. I do not have major concerns, and I would be happy to recommend the publication of the manuscript to facilitate the communication within the community.

But I do have some points that I want the authors to adjust or clarify, as follows:

1, Why the authors chose to determine the cryo-EM structure of the I551A variant of TMEM16A, rather than the I641A variant? Since the I641A variant seems to have the most significant effect, I am curious why this variant was not the choice by the author for structural determination.

2, In line 218 and 219, the authors state “Collectively, our functional characterization thus defines the importance of hydrophobic interactions within the isoleucine triad”. The “hydrophobic interactions” should be changed, because the I550, I551, and I641 are not within a distance for hydrophobic interaction in the WT structure. And the authors also mention in the abstract that these residues are not in physical contact.

3, To be self-consistent, the format when referring to a reference should be kept the same. In line 293 (ref. 12), line 364 (ref. 34), line 365 (ref. 35), and line 371 (ref. 37), the references should appear as superscript, as is done throughout the manuscript.

We thank all reviewers for their generally positive and constructive comments, which we have incorporated in our revision and which we have addressed in detail below.

Reviewer #1 (Remarks to the Author):

Gating the pore of the calcium-activated chloride channel TMEM16A

The calcium-activated chloride channel TMEM16A is an important channel from a physiological perspective. The channel protein is distinct from other families of channels both in terms of its three dimensional structure and in terms of its mechanisms. In this nice study, the authors address the molecular mechanisms of “gating”, by which the ion pore permits chloride conduction when a gate is open and prevents ion permeation when a gate is closed. The authors used scanning mutagenesis to identify amino acids that affect the equilibrium between the open and closed conformations and identify three hydrophobic residues (Ile 550, Ile 551, Ile 641) in particular that when mutated to alanine bias a conductive (open) form of the channel. These three amino acids contribute to the walls of the putative ion conduction pore. Structural studies, including cryo-EM structures of one of the mutants in both the calcium-bound and calcium-free conformations, provide insights into the calcium-dependent activation mechanism and the effects that calcium binding has on the pore of the channel. The authors propose that these three amino acids function as major constituents of the gate of the channel. The studies are well executed and technically sound. The studies provide new insights into the molecular mechanisms of gating in this family of channels. The combination of structural biology with rigorous electrophysiological approaches provides a rare depth of mechanistic insight into ion channel function.

The following specific comments are meant to help the authors improve their manuscript. I think it would be helpful for the authors to more clearly define what they mean by “gating” and what is meant by a “gate”. This comment relates to several specific comments below. Some of these comments are clarified by the accompanying manuscript, but it would make this manuscript stronger to address them here.

Throughout the manuscript we refer to ‘gating’ as the process that controls the opening and closing of an ion conduction pore, and the ‘gate’ as the location of residues contributing to the presumed highest energy barrier for ion conduction in the closed state. We have made further clarification as detailed in the responses to the specific comments below.

The authors note, in lines 60-62, “Despite the described evidence of a gate in the constricted pore region, the exact location of residues that obstruct ion flow in the closed conformation and the detailed spatial rearrangements during activation have remained elusive.” In the structures of Ca-bound and Ca-free TMEM16A, determined by the authors in this study and previous ones, the amino acids identified (I550, I551, I641) do not seem to undergo conformational changes and the dimensions of the regions of the pore that they surround do not seem to change much either. The authors may wish to clarify to the reader why their data supports the conclusion that

this region operates as the “gate” if it doesn’t seem to undergo conformational changes in these structures.

Although the structure of the Ca²⁺-bound TMEM16A might not show the full extent of conformational changes leading to pore opening, we think that it bears critical features of a conducting state and that its relation to the Ca²⁺-free structure shows major rearrangements underlying channel activation. Additionally, the combination of structural information and functional data provide further evidence for the location of a gate at the intracellular entrance of the neck:

- Compared to its Ca²⁺-free equivalent, the Ca²⁺-bound structure shows several features of an activated channel, which includes the conformational change of helix $\alpha 6$ upon interaction with the bound Ca²⁺ and a concomitant moderate increase of the pore size (which was described previously, see reference 12).
- We do not expect the expansion of the neck to be large. From a previous functional study that was based on cysteine accessibility experiments (reference 12), we know that the neck region remains narrow in the open state. This study showed that the intracellular vestibule at the entrance to the neck is accessible to small MTS reagents irrespective of the presence of Ca²⁺, thus demonstrating that this region is unlikely to limit the access of ions to the pore in the closed state. The same study also showed that a residue located only one helix turn above the accessible position is inaccessible in both Ca²⁺-free and Ca²⁺-bound conditions thus emphasizing that this region appears to constrict the pore even in the open state (reference 12).
- Our extensive mutagenesis study described here localizes residues whose sidechain truncation stabilizes the conducting state to the intracellular part of the neck (Figs. 1 and 2c).
- Finally, we find that the increase of the side-chain volume of one the three gate residues (I641), which is located farthest inside the neck, increases the barrier for conduction (as evidenced by the outward rectification of currents, Fig. 6b,c) thus providing further evidence for a constricted neck region even in the conducting state.

At the end of the first sub-chapter of the results (line 119-128), we have already described our view of the role of the three isoleucines in stabilizing non-conducting conformations and providing physical barriers that impede ion conduction in the closed state. In our revision we have now added the following sentence (line 128-130) to establish a relationship to known structures (a more detailed discussion on the structures is provided in the following sub-chapter):

‘While a moderate widening of this region upon Ca²⁺ binding was already found in cryo-EM structures of the protein¹², the functional data presented here imply a possible further expansion of the pore to be fully conductive.’

Line 111-113 – please provide reference for this statement.

We have added references 12 and 20.

Line 114. Is “outward rectification” the proper terminology for the small current seen at -100mV? The channels still seem to be inward rectifiers.

The sentence at line 114 refers to the current-voltage relationships of mutants in the Ca²⁺-bound state, where currents are large in both directions. In this case the mutations cause a moderate deviation from linearity which results in inward rectification in I550A and I641A, and outward rectification in case of mutants I551A and Q649A (outward rectification refers to relatively larger outward currents and inward rectification to relatively larger inward currents).

The small currents at negative voltage referred to by the reviewer shows basal activity under Ca²⁺-free conditions where currents are strongly outwardly rectifying as a consequence of the electrostatic barrier at the vacant Ca²⁺-binding site (in all cases we plot instantaneous currents and the displayed I-V relations thus reflect a property of permeation and not changes in the open probability).

Line 120, phrase “which show the strongest energetic contribution” was a bit confusing/unclear. I assume you are referring to the greatest shift in the Ca-response curve. It would be helpful for the reader to clarify the connection that you are making between free energy differences and Ca-activation curves.

We have described the relationship between the Ca²⁺ potency and the relative stability of closed and open states in the text (line 81-91) and the supplementary note. Additionally, we have added the following sentence (line 123-125):

‘...which show the strongest energetic contribution (i.e. the most pronounced left shifts in the concentration-response relations and the appearance of basal activity), are located between the inner part of the neck and the intracellular vestibule (Figs. 1d and 2a)’

Minor point, on several occasions, the authors refer to mutations as stabilizing the open or closed states (e.g. line 97). It seems possible that a given mutation that biases the open state may do so by either energetically stabilizing the open state or by destabilizing the closed state. Or vice versa for a mutation that biases a closed state.

We have now mentioned explicitly at the beginning of our result section that a left-shift indicates a relative stabilization of the open and a right-shift a relative stabilization of the closed state (line 86-91). Later we have used these terms interchangeably throughout the text as it is probably obvious to the reader that these are relative terms.

We have introduced the following changes to our manuscript (line 79-82):

‘We reasoned that residues contributing to a gate would face the pore and that truncation of their sidechains would increase the relative stability of conducting compared to non-conducting conformations of the channel. Such stabilization of an open state (or destabilization of a closed state) should be reflected in a change of the Ca²⁺ potency, ...’

Line 121-123, I found the phrase “latter observation” a bit unclear. Also, regarding that sentence: “The latter observation is consistent with the presence of bulky hydrophobic residues at the intracellular entrance to the neck functioning as steric and hydrophobic barriers that prevent ion conduction in the closed state of the channel.” I agree. However, the authors should discuss this hypothesis in the context of the observation that the conformations of the residues (I550, I551, I641) do not seem to change between the Ca-bound and Ca-free structures. If the dimensions of the pore at I550, I551, I641 do not change, how do the authors envision that they function as a gate? (This is the main question addressed in the accompanying manuscript, but it would help the reader to provide more context here.)

As mentioned before, although we observe a small expansion of the gate region, the detected change might not reflect the full opening of the pore in the conducting state. Our functional experiments that characterize the influence of the sidechain volume of the gate residue on conductance is consistent with a further expansion of this region.

We have reworded the term and added a sentence (line 125-130):

‘The observed effect of mutating these isoleucines is consistent with the presence of bulky hydrophobic residues at the intracellular entrance to the neck functioning as steric and hydrophobic barriers that prevent ion conduction in the closed state of the channel. While a moderate widening of this region upon Ca²⁺ binding was already found in cryo-EM structures of the protein¹², the functional data presented here imply a possible further expansion of the pore to be fully conductive.’

Line 182-185, “In these experiments, the stability of the open state shows an inverse correlation with the number of methyl groups within the isoleucine triad (Fig. 4a,b), further supporting the role of these residues as being part of a hydrophobic gate that excludes water and ions in the closed conformation.” The fact that mutations to the I550, I551, I641 residues biases the open state does not seem to necessarily indicate that these residues are part of the variable constriction in the channel that functions as a gate. Certainly their location relative to the putative ion conduction pore suggests that they may function in this regard, but it seems possible that their mutation favors an open conformation (e.g. through an allosteric mechanism) without necessarily indicating that the residues are part of the hydrophobic constriction that functions as a gate.

We agree that in light of the possibility of allosteric interactions between different parts of the protein these results cannot be interpreted unambiguously without prior structural information. However, since in the structure these residues enclose a focused region of the pore, they carry appropriate physicochemical properties expected for residues contributing to ion channel gates and they exert a pronounced energetic influence on the closed-open equilibrium, we think that the interpretation that the three isoleucines collectively function as a gate region is reasonable and reflects the most probable scenario. It should also be emphasized that the narrow neck constitutes a confined part of the pore whose residues have been mutated systematically in our study and no

other residues with expected properties have been identified as potential alternative candidates of a gate.

We have modified a sentence (line 199-202):

‘In these experiments, the stability of the open state shows an inverse correlation with the number of methyl groups within the isoleucine triad (Fig. 4a,b) that encloses the pore, further supporting the role of these residues as being part of a hydrophobic gate that excludes water and ions in the closed conformation.’

Line 191-193. A reference is needed.

We have added reference 29.

Line 200-202. More information regarding what the authors mean by “fractional contribution of the hydration energy” would help readers not familiar with this terminology/methodology.

We have added in line (line 218-222):

‘From an analysis similar to the one used for their truncation and under the assumption that the energetic effect of mutations is proportional to the hydration energy of sidechains³⁰, the fractional contribution (*i.e.* the proportionality constant) of substituted residues in stabilizing the open state was estimated to be 0.37 ± 0.11 .’

Line 227 – reference needed.

We have added a reference 11.

Lines 234-239: “In contrast, equivalent mutations of Ile 550 and Ile 551, which are located on the opposing helix $\alpha 4$, do not lead to strong rectification (Fig. 6d), suggesting that these residues do not contribute to rate-limiting energy barriers for conduction in the open state (Fig. 237 6e), and that they might instead have retracted from the pore.” The authors may wish to clarify what they mean by “retracted from the pore” since the conformations of Ile 550 and Ile 551 are very similar between the Ca-bound and Ca-free conformations of the channel. Perhaps they are hinting that the Ca-bound structure is not fully open?

Our results suggest that both Ile 550 and 551 do not directly contribute to the highest energy barrier for permeation in the conducting state since the increase of the sidechain volume does not show a size-dependent effect on conduction. The fact that in the structure of the Ca²⁺-bound protein both residues are still close to the channel constriction indicates that this structure might not represent a fully conducting state and that the residues would further retract from the pore constriction in the open state. However, the extent of the retraction is unclear and might as well be moderate as reflected in the inaccessibility of proximal residues for chemical modification as described before.

We have changed the text and added a sentence (line 257-259):

‘Instead, they might have retracted further from the pore constriction than observed in the Ca²⁺-bound conformation of TMEM16A, corroborating with a plausibly more extended rearrangement of the pore in a conducting state.’

Line 261-264, I did not understand the phrase “the gate dissociates”. (Again, this is clarified by the accompanying manuscript, but context would help here.)

We have reworded the sentences (line 283-287):

‘In the open state, the hydrophobic interactions that exclude the access of water to the gate region break down leading to the opening of a water-accessible path. As a result, the relative role of the three residues on the ion permeation path have changed as illustrated by the distinct effects of increasing sidechain volume on conduction, where mutations of Ile 641 but not of Ile 550 and Ile 551 severely perturb current-voltage relationships (Fig. 6).’

Line 300: “In summary, our study has identified a gate region that stabilizes the closed pore of TMEM16A ...” I agree that the studies of the I550, I551, I641 residues suggest that this region stabilizes a closed (non-conductive) conformation of the channel. In line with the preceding comments, the authors may wish to clarify what they mean by “gate region” and how the gate operates (does it change in dimensions, for example, to allow or prevent ion conduction?). As this is the subject of their accompanying manuscript, they should at least mention this.

We think that the gate prevents ion conduction in the closed state by narrowing the pore to a size that excludes water and thus prevents ion conduction. Upon pore opening our data suggest that the residues constituting the gate move apart to increase the pore size sufficiently to allow access of

We have rewritten the mentioned paragraph to make this clearer (line 325-332):

‘In the closed state, the proximity of hydrophobic residues at the intracellular entrance of the narrow neck prevents access of water and ions to this location. Upon activation, the breakdown of hydrophobic interactions in the gate region in conjunction with further conformational rearrangements of the protein, lead to the expansion of the constricting neck region in the anion conducting state. The structures presented here likely delineate a general mechanism for activation where the sequential rearrangement of the intracellular half of $\alpha 6$ couples to the narrow neck region of the pore to open the gate although they likely do not show the full extent of activation.’

In Figures 1d and 6b, it would be helpful to show the wild type I-V curves as well.

The reviewer refers to panels 2d and 6b. We have now shown the wild-type I-V curve in Fig. 2d and 6b as black dashed line (which partly overlaps with some traces of mutants in 6d). Due to the absence of basal currents in WT, I-V relationships for WT are only shown for the Ca²⁺-bound state.

The authors should comment on the absence/presence of density for PIP2 (diC8-PI(4,5)P2).

We have added the sentence (line 144-148):

‘Despite the presence of diC8-PI(4,5)P₂ in the sample, no densities that could be confidently attributed to the lipid analogue were found, which could either reflect a weakening of PI(4,5)P₂ binding to the channel in a detergent environment or alternatively be a consequence of its intrinsic mobility and the limited resolution of the data.’

Reviewer #2 (Remarks to the Author):

The present manuscript investigates the gating mechanism of the TMEM16A calcium activated chloride channel. These channels are key players in human physiology as they regulate multiple processes, from smooth muscle excitability to epithelial salt reabsorption and fertilization. Despite intense structural and functional scrutiny, the molecular mechanisms underlying the opening and closing of these channels remains poorly understood, as all current structures report on the closed conformation of the channel. Past work suggested that ion permeation through the TMEM16A channel is regulated by the electrostatic profile of the pore and by the release of a hydrophobic constriction along the pore. Here, Lam and colleagues use functional mutagenesis and electrophysiology to probe the roles of pore-lining side chains and show that mutations at the hydrophobic gasket specifically facilitate channel activation by increasing the EC₅₀ for Ca²⁺ and increasing the basal open probability in the absence of Ca²⁺. The structure of the I551A mutant in Ca²⁺-free conformation shows rearrangements around the α 6 helix that forms the Ca²⁺ binding site, while that of the mutant in the Ca²⁺-bound state is similar to that of the WT protein in the same conditions. Mutant cycle analysis shows that the residues forming the hydrophobic gasket are energetically coupled, and that the hydrophobicity of the side chains at these positions stabilizes the closed conformation of the pore.

Overall, this is an elegant and well-designed study that provides interesting insights into the opening mechanism of the TMEM16A channel. I have some requests for clarifications from the authors and would suggest they tone down some of their conclusions, that in some places seem to be a bit overstated relative to the data.

1) Did the authors account for the desensitization of the TMEM16A channel during their electrophysiological experiments? Do any of the mutants affect this process? Since the channel’s pore appears to be in a non-conductive conformation in all structures (presented here and in previous work), how can the authors rule out the possibility that they all represent a desensitized conformation?

In our experiments, current rundown was accounted for with a recording protocol that we have used in a previous study (reference 12). As displayed in the figure below (Response Fig. 1), the current rundown of WT proceeds with a time course in the order of 100s of seconds. The same figure shows that the time course of rundown changes in the three gate mutants. However, even in case of strongly increased rundown kinetics of the mutant I550A, which accelerates by a factor of about 10, it is still several orders of magnitude slower than the kinetics of the here described activation steps. A similar dependence of mutations on the rundown process has been observed by

Le et al 2019. As this process, which might be related to the dissociation of the lipid PIP2 from the channel, is not the focus of our study, we refrained from making direct comparisons in our manuscript.

With respect to the effect of inactivation on our structural studies, we have addressed the issue of a possible desensitization in the observed protein conformations in the comments below.

Response Fig. 1: Time dependence of rundown during the recording of Ca^{2+} -concentration response relationships. The displayed normalized currents show the current response of test pulses at saturating Ca^{2+} concentration that was recorded after each Ca^{2+} concentration step during the recording protocol. Data are averages of 7-10 recordings, errors are SEM. In panels displaying data of mutants, WT is shown as dashed line for comparison.

2) The authors present 3 new structures of TMEM16A: the WT channel in the presence of PIP2, and of the I551A mutant in Ca^{2+} -bound and Ca^{2+} -free states. Unfortunately, the insights that can be obtained from these 3 structures are limited: addition of PIP2 seems to do nothing, the I551A mutant has no effect in Ca^{2+} bound state and induces a small rearrangement of $\alpha 6$ in the Ca^{2+} free state. The structures are over-interpreted.

Although the WT structure in the presence of Ca^{2+} is similar to previously reported data, we disagree that the three structures would only provide limited insight into the observed functional properties. This is particularly the case for the mutant I551A, where the structure of the Ca^{2+} -bound state closely resembles WT, in line with electrophysiological data of the Ca^{2+} -bound state displayed in Fig. 2d. In contrast, the structure of the Ca^{2+} -free state of the mutant shows an important difference compared to WT in the conformation of α -helix 6, the key player in Ca^{2+} -activation, which in I551A has moved towards the Ca^{2+} -binding site. Although at first glance this change might appear minor, we have previously demonstrated the importance of the conformational rearrangement of $\alpha 6$ during the activation process (reference 12). The fact that the change observed here is the consequence of a mutation of a site in a different helix ($\alpha 4$), emphasizes the described coupling between both helices during the activation process. However, since the structures might not display fully activated conformations, we have toned down our interpretation.

- There is no evidence that the PIP₂ structure represents an intermediate on the way to opening as the authors claim (line 138-139). There is no evidence that PIP₂ binds to the detergent-solubilized channel protein, so that the protein could be in a desensitized state. Addition of PIP₂ to the solution does not mean that it will bind to the protein, it is possible that PIP₂ binding is weakened in the absence of a lipid membrane. To address this concern a structure of TMEM16A in a nanodisc supplemented with PIP₂ would be needed. However, I realize that this might not be a straightforward or feasible experiment. Nonetheless, the authors should tone down their statements on what the state represents and discuss the above-mentioned potential pitfall.

We have previously shown that the reconstitution of detergent purified protein in the absence of Ca²⁺ into proteoliposomes results in functional channels that mediate Cl⁻ flux in response to Ca²⁺ activation (reference 12). Here we proposed the relationship between the Ca²⁺-bound TMEM16A structure and a potential intermediate state on the pathway towards activation based on the fact that the brief exposure of TMEM16A to Ca²⁺, which was purified in the absence of Ca²⁺, was sufficient to promote Ca²⁺ binding and the concomitant rearrangement of $\alpha 6$ from its detached conformation observed in the Ca²⁺-free structure to a tightly bound conformation in the Ca²⁺-bound structure. As this conformational change is accompanied by a small expansion of the pore around the here identified gate region (which, although clearly detectable, might be insufficient to widen the pore to allow ion permeation), we attributed the conformation to a potential pre-open intermediate that was detected functionally in our accompanying study (reference 27). However, it is true that we do not have evidence of PIP₂ binding in our structures and we cannot exclude that the observed conformation might be biased by the detergent and might show a closer resemblance to an inactivated state that is stable upon dissociation of PIP₂.

We have added several sentences (line 144-156) to discuss potential pitfalls in the interpretation of the structures:

‘Despite the presence of diC8-PI(4,5)P₂ in the sample, no densities could be confidently attributed to the lipid analogue, which could either reflect a weakening of PI(4,5)P₂ binding to the channel in a detergent environment or, alternatively, be a consequence of its intrinsic mobility which impedes its identification at the observed resolution of the data. Since the purified protein conducts anions after liposome reconstitution¹² and undergoes structural rearrangements that are characteristic of activation within seconds of exposure to Ca²⁺, the vitrified protein likely displays a conformation that is functionally relevant. Still, since at its constriction the diameter of the pore is narrower than the size of permeating anions, its full opening might have been precluded in a detergent environment. Due to the potentially incomplete pore opening, the observation that $\alpha 6$ adopts an activated conformation suggests that the protein might display a pre-open intermediate (i.e. a Ca²⁺-activated non-conducting state) that we describe in an accompanying manuscript²⁷, although we cannot exclude a closer resemblance to an inactivated state that is adopted upon dissociation of PI(4,5)P₂.’

- The authors repeatedly refer to the I551A mutant as constitutively active (i.e. lines 140, 146, Fig. 3 legend). However, the authors data are not consistent with this claim: the I551A mutant

is Ca²⁺ and voltage dependent. I think it is more appropriate to refer to this mutant as “activating”, as the authors do in other portions of the manuscript (i.e. in the abstract).

We would like to emphasize that the voltage dependence of basal currents (in the absence of Ca²⁺) is a property of permeation and not of gating. I551A is amongst the mutants that display the most shifted EC₅₀ and concomitant basal activity. These basal currents show a strong outward rectification, which does not reflect a voltage-dependent change in the open probability but is instead a property of permeation (the IV relationship is plotted from instantaneous currents after a voltage jump and the currents do not show a relaxation component that can be attributed to conformational changes in the protein). As we described previously and now more explicitly explain in the appendix (Supplementary Note), the observed Ca²⁺ dependence is due to the transition from a low-conductance open state in the absence of Ca²⁺ to a high-conductance open state (Appendix Fig. 1C). We have previously characterized these open states in detail and found that this phenomenon originates from direct coulombic interactions between the bound Ca²⁺ ions with the permeating anions (reference 20). Since the binding site is in close proximity to the pore and is highly negatively charged in the apo open state, the energy barrier is high for anion conduction and the conductance is thus low. When two Ca²⁺ ions are bound, the negative potential at the binding site becomes neutralized and the electrostatic barriers in the pore are alleviated, allowing anions to diffuse with a higher conductance. As such, even when the Po remains unaltered in the absence of Ca²⁺ and at saturating Ca²⁺ concentrations, the change in the conductance by Ca²⁺ would appear similar to an activation (Supplementary Note Fig. 1). The voltage dependence of the concentration-response relations is related to the Ca²⁺ binding site being situated within the transmembrane electric field. The observation of constitutive activity supports the notion that the channel samples the open state in the absence of Ca²⁺. It is worth noting that the mean current response of a single channel is given by $\sum_j i_j P_o_j$. In fact, one can qualitatively deduce that the Po at zero Ca²⁺ for I551A is not much lower than at saturating Ca²⁺ concentrations (as determined in our accompanying study, reference 27) given that the conductance difference between the apo and the Ca²⁺-bound states (Fig. 2D) can account for much of the dynamic range of the concentration-response relation (Fig. 2B).

- I am not sure what mechanistic inferences can be drawn from the I551A mutant structures. The observation that the mutation affects the conformation of $\alpha 6$ is interesting and supports the notion this residue plays a role in gating. However, I am not sure how the authors can claim that “The relative stabilization of the open state in the mutant enables $\alpha 6$ to adopt a seemingly activated conformation...” (lines 164-165, a similar concept is posited in line 290). What supports the idea that $\alpha 6$ is in an activated conformation? Given that the pore is in a non-conductive conformation and that we do not know what the open TMEM16A pore looks like, this statement seems unwarranted.

In light of the relatively high open probability of the mutant I551A in the absence of Ca²⁺ (see arguments above), we assume that its structure bears features of a basally active state of the channel, irrespectively of whether the pore is in a wide-enough conformation to permit ion conduction. Since the transition from the apo state (PDB: 5OYG) to the Ca²⁺-bound state (PDB: 5OYB) involves the straightening of helix 6 and that this helix is in a partially straightened

conformation in the apo mutant structure, it is reasonable to be interpreted this as a feature that is characteristic of the basally active state of the channel.

The paragraph discussing the structure and the associated inferences should be drastically toned down.

We have introduced several changes that have toned down our structural discussion by including alternative interpretations.

- A minor concern is that the density of the $\alpha 6$ helix in the Ca^{2+} -free I551A mutant map is quite weak. I wonder how confident are the authors that there is a π - to α -helix transition in this mutant.

In the Ca^{2+} -bound state of both WT and I551A, the π -helical region where the i+5 pattern for backbone hydrogen bonding is apparent encompasses G644 to Q649. In the map of the Ca^{2+} -free structure the density of $\alpha 6$ is well-defined until Q649 with weaker density extending until N651. C-terminal to this position the helix is not defined due to its increased flexibility compared to the Ca^{2+} -bound state. Due to the clear definition of the position of Q649, the α -helical character of the region can be assigned with confidence.

- At what σ values are the densities shown in Extended Data Fig. 6?

The densities are shown at between threshold 0.02-0.035 in ChimeraX, which correspond to sigma values between 4 and 5.

3) The authors use mutant cycle analysis to show that the positions around the hydrophobic constriction are functionally coupled. They then use a Markov model to extract the energetic contributions of individual methyl groups in the pore to the EC50 for Ca^{2+} and to hydration energy of the pore. The use of this model is central to the present work, as it enables the authors to extract detailed, quantitative mechanistic information on the opening process. Overall, this analysis is very elegant and intriguing. However, the description of the Markov model in the main text is non-existent and is very succinct in the Methods section. As such, some of the non-obvious insights are cryptic (i.e. why does the EC50 first decrease with decreasing number of methyl groups and then saturates?) and it is very difficult to evaluate the robustness of the estimation of the parameters of the model. A thorough description of the model and of how parameters are estimated is needed. Further, the authors should document how robust the fitting process is (i.e. how coupled are the various parameters in the model? how unique is the choice of parameter set for each condition within the parameter space of the model?). I suggest the authors add an appendix to the manuscript where these critical issues are thoroughly discussed, and the model and its assumptions are validated. Without this documentation, I have concerns that the estimates of the modeled gating parameters might not be sufficiently precise and

uniquely determined to enable the quantitative inferences on the energetic coupling and of the contributions of individual side chains to the gating process.

We have now added an appendix (Supplementary Note) that more thoroughly examines and discusses some of the less obvious properties of the model and the errors in parameter estimation. We hope that this would increase the clarity of our analysis and improve the manuscript as a whole.

Reviewer #3 (Remarks to the Author):

Calcium-activated chloride channels are transmembrane proteins that play important roles in various physiological processes, such as neuronal signaling and smooth muscle contraction. TMEM16A encodes one such channel, whose activity is up-regulated upon Ca²⁺ binding, with different responding outcomes depending on the types of tissues where TMEM16A is expressed. Unraveling the detailed mechanism of channel activation and regulation will greatly improve our understanding of this type of proteins, and benefit development of therapeutic strategies targeting TMEM16A, given its involvement in certain diseases, such as cystic fibrosis. Structures of TMEM16A in its apo-form and Ca²⁺-bound form have been determined by cryo-electron microscopy in previous studies, revealing conformational alteration between the two states. In addition, a gating regulatory mechanism for the ion conductance was suggested. As a follow-up study, in this manuscript, Lam et al. carried out sophisticated electrophysiology experiments and extensive data analysis for systematic mutated variants of TMEM16A. The authors pinpoint amino acid residues that cause the stabilization of the closed conformation and the open conformation, respectively. Interestingly, a series of hydrophobic residues (I550, I551, and I641) were found to have a significant effect on stabilizing TMEM16A in its closed conformation. The authors attributed the effect of double or triple mutation of these three residues to their functional coupling. Moreover, cryo-EM structures of the I551A variant of TMEM16A were determined in the absence or presence of Ca²⁺, respectively, with the Ca²⁺-free I551A mutant showing similar conformation but noticeable difference. The experiments in this study were well executed and data analysis was performed carefully. The manuscript was very well written and easy to follow, and it was a delight to read the manuscript. The findings by the authors further our understanding of the regulatory mechanism of TMEM16A to a deeper extent. I do not have major concerns, and I would be happy to recommend the publication of the manuscript to facilitate the communication within the community.

But I do have some points that I want the authors to adjust or clarify, as follows:

1, Why the authors chose to determine the cryo-EM structure of the I551A variant of TMEM16A, rather than the I641A variant? Since the I641A variant seems to have the most significant effect, I am curious why this variant was not the choice by the author for structural determination.

We have chosen the mutation I551A for two reasons. One is related to its better expression properties compared to I641A, the second to its location on $\alpha 4$, which provides insight into the coupling between $\alpha 4$ and $\alpha 6$, latter of which is directly involved in Ca^{2+} binding. It should also be emphasized that the effect of I551A is qualitatively very similar to I641A with respect to the left-shift of the EC_{50} of Ca^{2+} and basal activity.

2, In line 218 and 219, the authors state “Collectively, our functional characterization thus defines the importance of hydrophobic interactions within the isoleucine triad”. The “hydrophobic interactions” should be changed, because the I550, I551, and I641 are not within a distance for hydrophobic interaction in the WT structure. And the authors also mention in the abstract that these residues are not in physical contact.

We agree that these residues are not in van der Waals distance. The description hydrophobic interaction is likely appropriate since water is very likely excluded in the closed state given such a confined geometry, as observed in previously published MD studies. Such gating mechanism has been referred to as hydrophobic gating, where in some channels residues of the gate region, although close, do not necessarily have to be in van der Waals contact to inhibit ion conduction in the closed state.

3, To be self-consistent, the format when referring to a reference should be kept the same. In line 293 (ref. 12), line 364 (ref. 34), line 365 (ref. 35), and line 371 (ref. 37), the references should appear as superscript, as is done throughout the manuscript.

These specific reference numbers were formatted according to the style of the journal in case the preceding character is a number.

Reviewer #1 (Remarks to the Author):

All of my comments have been addressed. It is a beautiful paper.

Reviewer #2 (Remarks to the Author):

The authors have done an excellent job of addressing all of my concerns and should be congratulated for an exciting contribution to our understanding of the TMEM16A gating mechanism.